# Decellularized Avian Cartilage, a Promising Alternative for Human Cartilage Tissue Regeneration

**DOI:** 10.3390/ma15051974

**Published:** 2022-03-07

**Authors:** Joseph Atia Ayariga, Hanxiao Huang, Derrick Dean

**Affiliations:** The Biomedical Engineering Program, College of Science, Technology, Engineering and Mathematics (C-STEM), Alabama State University, 1627 Hall Street, Montgomery, AL 36104, USA; jayariga7546@myasu.alasu.edu (J.A.A.); hhuang@alasu.edu (H.H.)

**Keywords:** chondrocytes, extracellular matrix, cartilage, decellularization, scaffold

## Abstract

Articular cartilage defects, and subsequent degeneration, are prevalent and account for the poor quality of life of most elderly persons; they are also one of the main predisposing factors to osteoarthritis. Articular cartilage is an avascular tissue and, thus, has limited capacity for healing and self-repair. Damage to the articular cartilage by trauma or pathological causes is irreversible. Many approaches to repair cartilage have been attempted with some potential; however, there is no consensus on any ideal therapy. Tissue engineering holds promise as an approach to regenerate damaged cartilage. Since cell adhesion is a critical step in tissue engineering, providing a 3D microenvironment that recapitulates the cartilage tissue is vital to inducing cartilage regeneration. Decellularized materials have emerged as promising scaffolds for tissue engineering, since this procedure produces scaffolds from native tissues that possess structural and chemical natures that are mimetic of the extracellular matrix (ECM) of the native tissue. In this work, we present, for the first time, a study of decellularized scaffolds, produced from avian articular cartilage (extracted from *Gallus Gallus domesticus*), reseeded with human chondrocytes, and we demonstrate for the first time that human chondrocytes survived, proliferated and interacted with the scaffolds. Morphological studies of the decellularized scaffolds revealed an interconnected, porous architecture, ideal for cell growth. Mechanical characterization showed that the decellularized scaffolds registered stiffness comparable to the native cartilage tissues. Cell growth inhibition and immunocytochemical analyses showed that the decellularized scaffolds are suitable for cartilage regeneration.

## 1. Introduction

The treatment of damaged cartilage is a big challenge due to the poor regenerative capacity of the dense avascular tissue. Current management practices include diet and exercise in mild conditions, capsaicin, lidocaine patches, as well as steroid administration in moderate conditions, intra-articular hyaluronic acid injection for similarly moderate conditions [1,2,3,4]. The employment of stem cell therapy has become popular among clinicians in efforts to regenerate cartilage [2,4,5,6]; however, the most popular practice has been the micro-fracture procedure, which recruits cells into the dense cartilage to initiate cartilage formation and regeneration [7]. Nonetheless, all these procedures have their drawbacks [8]. In extreme cases, joint replacements are the last resort, and this can also result in disability [9]. The surge in tissue engineering and its applications has become a great beacon of hope for clinicians who are grappling with the management of cartilage-related conditions [10,11,12,13]. Cartilage is a non-vascular tissue, consisting of chondrocytes, spatially dispersed throughout a compact extracellular matrix (ECM), which is capable of bearing loads [14]. The thick ECM of the cartilage tissue is accountable for the tissue’s mechanical properties. It is composed of a liquid phase (mainly of water and inorganic ions making up 65 to 80% of the wet weight), and the solid phase that consists, predominantly, of collagens (mainly type II collagen). Other constituents include proteoglycans, aggrecan, decorin, fibronectin, laminin, lipids, biglycan, hydroxyapatite and fibromodulin, which make up the fraction of the dry weight [15,16,17]. This tissue has a low cell count to area ratio, and its avascular character usually leads to inadequate self-repair and regenerative abilities after cartilage damage [18].

Cell-free tissue grafting procedures, carried out by Roessler et al., 2015, demonstrated that Col-1-based scaffolds, directly implanted into cartilage defects (in a press-fit manner), showed positive results for cartilage regeneration after a 4-year follow-up [19]; however, at 5-year follow-up, their data pointed to unsatisfactory outcomes [20]. In the case of extreme cartilage degeneration, knee replacement has always been an option. Research related to the assessment of the performance of artificial joints with Tresca stress, for which the use of in silico analysis of materials for metal-on-metal total hip arthroplasty with Tresca stresses, using a 2D axisymmetric finite element prediction model, was successfully demonstrated by Ammarullah et al., 2021 [21]. In a similar study, Jamari et al., 2021, demonstrated the effect of surface texturing as dimples on the wear evolution of total hip arthroplasty [22].

Antibiotics, growth hormones, and anthelminthic used for disease control and the growth of livestock have become a major concern, regarding the safety of livestock products [23]. The risk associated with these include some human illness, of which the etiology originates from the consumption and the use of these livestock products, such as the cartilage or bone for tissue transplants that are known to sequester residual antibiotics, growth hormones or these anthelminthic drugs. While xenografts containing these elements (if not screened out) might provoke immune reaction, and more adversely, also lead to other zoonotic disease infections and health concerns, the decellularization process is able to remove these residual hormones, drugs or antibiotics, hence, rendering the final construct free of residues of antibiotics, hormones or anthelminthic [24,25]. Thus, health risks associated with using cartilages from animals that have been treated with growth hormones or antibiotics are drastically reduced, if not fully eliminated, during the decellularization process.

The main objective of cartilage regeneration, therefore, is to produce a regenerated cartilage tissue that can be fully integrated into the articular surface and the subchondral bone. Additionally, the regenerated tissue should possess both the requisite mechanical and chemical properties matching that native tissue [26]. The extracellular matrix of cartilage presents a promising and better alternative biomaterial for cartilage tissue regeneration because they serve as scaffolds for the reseeded chondrocytes, as well as providing the requisite cues for the cells’ attachment, growth and development during the cartilage tissue regeneration. Several decellularized matrices of cartilages, such as from bovine, porcine and humans, have been shown to possess bioactivity [27,28]. Decellularized scaffolds have similarities in architectural features to the native tissue and, hence, similarity in the microenvironment in which reseeded cells can grow. Such recapitulation of the microenvironment could promote similar epigenetic cues [29,30,31], because of mechanical, molecular, and chemical functional groups of the extracellular matrix (ECM) microstructure.

Several animal tissues, including humans, have been studied for cartilage, bone, bladder, vein, and other tissue engineering [32,33,34,35,36,37]. Obtaining cartilage tissue from a human donor implies surgery that usually causes wounds and can lead to disability, since this tissue has poor regenerative capacity [38]. Sourcing cartilage tissues from cadavers is surrounded by moral and ethical concerns, as well as low availability too [39]. In the case of animal tissues, decellularized scaffolds from certain animals, such as porcine or bovine, are restricted by certain religions; for instance, while Islam prohibits the use of porcine tissues, Hinduism forbids the use of bovine tissues [40]. 

The use of avian tissue, however, has no recognized restrictions. For this reason, we were inspired to decellularize avian cartilage for human cartilage regeneration. The ease of producing decellularized avian cartilage scaffolds (DACS) and the availability of avian cartilage makes it a promising alternative for engineering human cartilage tissue. The decellularized construct is biodegradable, biocompatible and the production process has no biosafety concerns [41,42,43,44]. The alignment and orientation of collagen fiber in avian cartilage tissue are similar to that found in human cartilage [45,46]. More so, the structure of the articular cartilages of both humans and avians are similar, and consist of a superficial zone with horizontally aligned collagen fibers, predominantly collagen I, the transition zone with randomly aligned collagen II fibers, the deep zone with vertical collagen alignment [47]. This work is the first attempt at engineering human cartilage tissue using avian cartilage as a tissue source, and this is the first attempt at understanding the cellular and molecular interaction between the decellularized constructs and the reseeded human chondrocytes. 

In cartilage development, collagen II and GAG are critical in guiding cellular survival, attachment, proliferation, differentiation, and migration. These molecules also are known to be the significant components of the ECM of cartilage [48]. Hence, in cartilage regeneration, the intent is to have a neo-cartilage, consisting of high collagen II and GAG contents that are comparable to those of natural cartilage. Thus, in this work, we investigated the synthesis and mRNA expression of collagen by the chondrocytes. 

One major setback in tissue graft or organ transplant, from a donor of a different species from that of the recipient, is the possibility of zoonosis and immunological rejection [49,50,51,52]. The major concerns have been diseases, such as herpesviruses, retroviruses, Toxoplasma gondii [53,54], Mycobacterium tuberculosis, encephalomyocarditis virus [55]. However, these risks are screened out during decellularization and the immunogenic elements eliminated from the final decellularized construct.

Cytocompatibility studies of mesenchymal stem cells (MSCs) have been reported for decellularized bovine cartilage tissue for human cartilage regeneration [56]; in this study, we report similar studies using the decellularized avian cartilage. We also investigated the biophysical characteristics of the avian articular cartilage tissue and the mechanism by which decellularized scaffold geometric features affect human chondrocyte growth, attachment and morphology, and finally, the dynamic reciprocity of interactions between scaffold and chondrocytes. This study shows that DACS modifies the cell phenotype and stimulates chondrocyte cellular processes, relating to ECM formation and remodeling, hence, providing a good candidate for the tissue engineering of cartilages. The use of avian cartilage provides an exciting possibility of repurposing these commonly available biological tissues for cartilage tissue regeneration in the treatment of osteoarthritis and rheumatoid arthritis. 

## 2. Results

### 2.1. Extraction and Decellularization of Avian Articular Cartilage

The decellularization process was employed to reduce the immunogenic components of the cartilage tissue by eliminating the cellular elements, such as proteins, nucleic acid, and lipids, while maintaining the mechanical and bioactive properties of the cartilage ECM. Figure 1A–G shows the extraction and sterilization process for constructing the decellularized scaffolds, whereas Figure 2A–C depicts the scanning electron micrographs of the decellularized scaffolds at different magnifications. 

#### 2.1.1. Scanning Electron Microscopy

SEM images showed the morphological features of DACS, which consists of a sheet of tissue with numerous pores. The pores created by the elimination of the chondrocytes via decellularization, as shown in Figure 2A–C, provides porous and interconnected scaffolds for chondrocytes’ attachment and proliferation. Chondrocyte, therefore, can infiltrate, colonize and proliferate in the DACS. 

#### 2.1.2. Physical Characteristics of Decellularized Cartilages

The quantitative analysis of ECM thickness of old and young avian cartilage was conducted. The data in Figure 3A–D compares the two treatments: young and old avian articular cartilages. While the old avian articular cartilages registered a mean thickness of 15.54 µm, the young avian articular cartilages showed an average of 8.0 µm. Even though these results indicate great differences, it is important to point out that the dimensional analysis here is shown in averages, of which there is the presence of heterogeneity in dimensions. This heterogeneity in thickness and other macroscopic features, such as porosity, implies structural heterogeneity, thereby impacting the decellularized scaffold dynamic interactions with the seeded cells. These findings highlight, on the one hand, the significant differences in cartilage ECM formation and remodeling during growth and development, since older avian cartilages consisted of ECM thickness that was greater than the young ones, while on the other, it also indicates the limitations put on chondrocytes during cartilage remodeling, with the understanding that thicker ECM presents limitations to cell–cell signaling, communication, migration and recruitment and, hence, slows down cartilage repair after injury, or during the cartilage wound healing process. The thinner thickness observed in the young cartilage presents an ECM texture indicative of a high degree of porosity, which plays a vital role in higher cell–cell signaling, chondrocyte migration and, hence, cartilage tissue regeneration and remodeling. Understanding these differences also points to the difficulty for cartilage regeneration regarding older people. A high frequency of cartilage-related diseases and conditions have been associated with older people; hence, the aged with thicker cartilage ECM seemingly have their cartilage tissue barricade cell–cell contact, limiting the speed of cell–cell signaling processes. It is obvious from these data that the thickness of the older avian articular cartilage is double that of the younger avian articular cartilage, and these are significant features that are crucial in designing and fabricating biomimetic porous scaffolds that could possibly recapitulate the natural cartilage architectural features. A visual rendition of DACS (Figure 3B) shows the 3D rendition of the decellularized cartilage scaffold image used for the evaluation of pore size distribution and the estimation of porosity. As shown in Figure 3C, the old cartilage scaffolds had much thicker ECM compared to the young cartilages (Figure 3D). This implies a lesser porosity compared to the young cartilage scaffolds, and since porosity is directly related to void volume (Vv), as indicated in Figure 3B, we infer from these results that reseeded cells’ migration, invasion, and infiltration of the scaffolds will be much quicker in young avian articular cartilage’s scaffolds compared to the old ones. However, it is possible that the thicker ECM, as found in the old, will have a much better capacity to hold very vital bioactive molecules that can potentiate transcriptional regulations and cells growth and ECM remodeling. 

#### 2.1.3. Comparative Physical Analysis of Young and Old Avian Cartilage Tissue

A measure of chondrocyte densities, as shown in Figure 4A, was an indirect method of measuring the resulting pore distribution and densities, which are vital for reseeded cells’ migration, proliferation, mass flow, nutrient diffusion, and oxygen circulation, since the decellularization process will, in effect, remove these cells, leaving behind empty pores. In general, the overall mean density of the young avian articular cartilage chondrocytes was averaged at 6.5 × 10^−3^ chondrocytes per unit area, whereas the old avian articular cartilages registered 4.39 × 10^−3^ chondrocytes per unit area, indicating a 1.49-fold difference between the two treatment groups. Such a difference indicates the biological activity difference between the two sets, and the younger cartilage could have far more cellular events occurring than the older cartilage. For the number of chondrocyte counts per microscopic field in cartilage tissues, as indicated in Figure 4A, young avian articular cartilage showed the highest number of counts, with a mean count of 39.25, whereas the old avian articular cartilage showed a mean count of 24.62, reaffirming the differences in chondrocytes densities, as reported in the same Figure. As shown in Figure 4A, the young avian articular cartilages registered a higher number of isogenous groups (14.12), almost double that of the old avian articular cartilages, which recorded an average of 7.63, indicating a positive correlation between cell density and isogenous counts.

As shown in Figure 4B, there exist dissimilar mean isogenous group areas between young and old avian articular cartilages, with the young avian cartilages registering a mean area of 293.1 µm^2^ and the old avian cartilages holding a mean area of 251.7 µm^2^. However, when individual chondrocyte sizes were compared, a 61.0 µm^2^ average area was recorded for the old, a 20.7 µm^2^ higher than the young, which recorded an average area of 40.2 µm^2^. These data seem to point to two important things: A nascent isogenous group, which showed greater area and might be indicative of lesser ECM intrusion. In contrast, the old articular cartilages with smaller isogenous group areas imply that continuous ECM molecules’ deposition, remodeling, and thickening of the extracellular matrix may have an intrusive effect on the isogenous group area, thereby squeezing the isogenous group area into smaller and smaller spaces with time, as cartilage ECM thickens. The individual chondrocyte’s area, however, remained favorably higher in the old cartilages, implying that ECM thickening did not have any huge impact on the individual cell’s area. 

### 2.2. Protein Assessment of DACS

Various detergent-based protocols have been adapted for the eradication of tissue cells [57,58]. The SDS decellularization technique has been hailed as one of the most effective methods for removing cytoplasmic cytoskeletal proteins [59,60] and nuclear DNA [59]. This study compared the effectiveness of removing cellular proteins by SDS decellularization to that of Triton-X. As displayed in Figure 5A, total protein measurements were quantified and displayed using the SDS-PAGE gel. SDS-PAGE analysis of cellular proteins after 12-h and the 24-h time points of decellularization in 1% SDS showed that the intensities had decreased to background levels (Figure 5B).

The 1% Triton-X decellularization treatment saw a smaller decrease in mean densitometric value, registered a mean densitometric value of 4.23 at the 3-h time point. This represented a difference from the untreated sample control native tissue of 1.87. At the 24-h time point, 1% Triton-X gave a densitometric mean of 1.97. In addition, the SDS treatment demonstrated a faster removal of cellular protein than the Triton-X treatment, and it was observed that the ECM of the SDS decellularized cartilage received significantly less damage than for the Triton-X treatment.

### 2.3. Nucleic Acid Assessment of DACS

SDS treatment of our scaffold eradicated the nucleic acid content of the decellularized tissue at the 36-h time point, as shown in Figure 6A. The 12-h time point, however, shows a faint band, displaying the presence of residual DNA. The DNA content of decellularized cartilage tissue after Triton-X treatment demonstrated the complete removal of nucleic acid at 12-h and 36-h (Figure 6A). In Figure 6B, the quantitative analysis of the nucleic acid content revealed a densitometric mean value of 2.06 for the native samples. This value dropped to 0.03 at the 12-h point for SDS treatment and below background values for all the Triton-X treatments. In light of these results, while Liu et al. indicated that the SDS treatment was the best method for removing nuclear DNA in the porcine aortic valve [61], our study indicated that Triton-X treatment was faster at removing DNA in avian articular cartilage than the SDS treatment.

### 2.4. Analysis of Lipid Content of DACS 

In this study, as indicated in Figure 7A, the effectiveness of the decellularization protocols on eliminating cartilage tissue lipid was investigated to ensure that the decellularized scaffolds were devoid of immunogenic molecules. The native tissue registered a mean of 0.84% wet weight. We found that 3 h of decellularization with SDS brought down the lipid content to 0.39%, and this value fell to 0.04% at 36 h of SDS treatment.

### 2.5. Analysis of Sulfated Sgags Content of DACS 

In this study, as depicted in Figure 7B, we investigated the effects of SDS decellularization on the sGAG content of the decellularized cartilage. SDS decellularization for 3 h produced an average sGAG weight of 153.30 µg/100 mg of wet cartilage sample, while the 36-h decellularization gave an average mean sGAG weight of 130.26 µg/100 mg of wet cartilage sample. The native cartilage control recorded 360.27 µg/100 mg of wet cartilage sample. Since the measure of sGAG content is an indirect measure of ECM integrity [62], it is evident from this study that decellularization via SDS performed better in maintaining the structural integrity of the resulting decellularized scaffold, since a high amount of sGAGs were still available after 36 h of decellularization. 

### 2.6. Histochemical Analysis

#### 2.6.1. DACS Lipid Content

Lipid affects the immune reaction during transplantation [63]. Histological staining with oil red stain was used to characterize the effect of decellularization on lipid content. Complete removal of the oil red stain (red color), indicative of the complete elimination of lipids, occurred after 36 h of decellularization, as shown in Figure 7C. Also, Figure 7B showed that lipids were absent after the 24- and 36-h’ time points. These are positive indicators of eliminating immunogenic lipids and making the resulting scaffold biocompatible for tissue culture and engineering purposes.

#### 2.6.2. DACS Sulfated Gags (Sgags) Content

The effects of decellularization on DACS glycosaminoglycans were assessed by assessing the amount of sGAGs present in the DACS constructs. The decellularized tissues at various set time points of decellularization were stained with Alcian blue stain, which is specific for sGAGs. As depicted in Figure 7B, at the 0-time point, SDS treatment gave a significantly pronounced blue color, indicative of the high sGAG content of the tissue. However, a gradual decline in the intensity of the blue color with increasing decellularization time was observed.

### 2.7. Biomechanical Characteristics of DACS

The modulus (stiffness) of the scaffolds was correlated with the scaffold morphology. The mean modulus was determined in three replicate measurements and used to plot a bar chart, as shown in Figure 8. The native cartilage exhibited a modulus of 292 Pa, while the decellularized cartilage exhibited a modulus of 201 Pa. The modulus of the decellularized scaffolds agrees with its higher porosity.

### 2.8. Chondrocyte Viabilities on Monolayer Plate and on DACS

As shown in Figure 9A–D, in the first 15 days of culture, the phase-contrast micrographs of cells showed continued growth, both in the monolayer plate and in DACS. In Figure 9E, viability of the chondrocytes assessment via alkaline phosphatase assay indicated an increased proliferation with increasing days of culturing.

### 2.9. Alkaline Phosphatase Activity Analysis of Chondrocyte Lysates from Monolayer Control Plates and Cells Seeded on DACS

The tissue-nonspecific alkaline phosphatase (ALP) is a known marker of both chondrocytes and osteoblasts [61,62,63,64,65]. It has also been identified on the membranes of matrix vesicles [66,67]. In this work, cell proliferation was measured by assaying for p-nitrophenol phosphate conversion via alkaline phosphatase. As shown in Figure 9E, the control monolayer plate group exhibited a faster rise in alkaline phosphatase activity. On day 10, it peaked at a mean of 3.22 U/mg of protein and decreased to a mean value of 2.36 U/mg on day 15. The DACS-seeded chondrocytes showed a gradual linear increase in ALP activity and a less steep slope than the control group, indicative of a lower unit conversion of p-nitrophenol phosphate on all culturing days. The highest ALP activity of chondrocytes on DACS was recorded on day 15, with a 2.12 U/mg of protein lysate, demonstrating that increasing culturing days, correspondingly, gave an increased production of ALP, which is indicative of cartilage tissue regeneration and chondrocytes’ proliferation. 

### 2.10. Phenotypic Characteristics of Chondrocytes Seeded on DACS

There was an observed phenotypic difference between chondrocytes plated on the six-well plate control and those seeded on DACS. Among the most obvious observation is the compliant and elongated morphology of the DACS-seeded chondrocytes. Cells located between the plate surface and the scaffold boundaries, as shown in Figure 10A, seemed to show a general spear-like morphology, with a long “proboscis”, spearing into the DACS scaffold and leaving most of their cytoplasmic body outside the scaffold. This observation was seen in phase contrast microphotographs, after the first 2 to 4 days of culturing. On day 3 of culturing, chondrocytes appear to be weaving through the DACS scaffold pores, as shown in Figure 10B. A cursory plausible answer to why such shape morphing occurs at these early stages of culturing could be that chondrocytes extend their pseudopodia into the DACS scaffold to probe the scaffold, while leaving their main body outside the scaffold to ascertain the biocompatibility of the construct. We also propose that the thin, stretchy phenotypic appearance is likely due to the smaller pore sizes of the reseeded chondrocytes. This structure of the scaffold means that chondrocytes are forced to morph their shape to fit the tiny pores, in order to gain access to the scaffold interior for invasion, infiltration, and proliferation. Day 15 of culturing chondrocytes on the DACS scaffold, as shown in Figure 10C, depicts cells complete invasion and infiltration of the scaffold, to the degree that the entire scaffold was covered with chondrocytes. Figure 10D shows the immunofluorescent staining of the DACS scaffold, with DAPI at day 5, where cells were shown to have colonized the scaffold. Figure 10E shows the actin cytoskeleton in green, depicting stressed fibers and an indication of strong attachment to scaffolds. Figure 10F shows the merge of the DAPI and phalloidin. 

There were observed differences in cell morphology and proliferation between cells seeded on the DACS scaffolds and the monolayer plate control; these differences might have arisen due to several factors, among which include matrix mechanics, fluid transport, scaffold/substrate stiffness, presence/absence of localized inductive molecules, etc. Reports of the scaffold effect on cell morphology, and proliferation due to scaffold properties, has been reported elsewhere [68,69,70]. In this study, we observed that the monolayer plate control cells exhibited lower expression of adhesion molecules in comparison to the cells cultured on the 3D DACS scaffold (data not shown); similar observations have been reported elsewhere [71]. The fundamental interactions between cells and scaffolds, in light of the gene expression during tissue formation, remain a significant aspect of tissue engineering study [72,73]. Scaffolds stimulate cellular protein expressions that also potentiate the activation of autocrine and paracrine signaling cascades that are critical in tissue development in vivo, and cell shape and morphology in vitro [74]. In some tissue engineering approaches, the focus has been on delivering growth factors, e.g., IGF, FGF-2, VEGF, BMP, to cultured cells to induce specific cellular differentiation or cell signal secretion in the micro-environment [75,76]. For instance, VEGF-A has been noted to enhance both angiogenesis and wound healing [77]. However, in this work, no growth factor was added.

Another significant observation was the bidirectional morphological phenotypes observed in cells grown on the DACS scaffold, especially at the early culturing days, compared to the flattened and almost isogenic phenotypes observed in the plate control, which stood as sharp contrasts. It is interesting to note that similar phenotypes have been recorded before in the literature, in which a scaffold’s 3D matrix has been demonstrated to play a fundamental instructive role in cell polarity [78,79].

### 2.11. Evaluation of Cytotoxicity of DACS Scaffold at Different Time Points of Decellularization

The ECM comprises a non-cellular three-dimensional macromolecular network of collagens, proteoglycans, fibronectin, laminin, and other glycoproteins [80,81,82]. Most decellularization aims to remove immunogenic agents [83] in the form of cellular components, since these immunogenic agents are the activators of the host immune system and can be destructive during tissue transplantation [84]. Therefore, a constructive remodeling of the scaffold by reseeded cells occurs if the scaffold is wholly decellularized and contains no endotoxins, immunogens, or contaminants that can affect the surrounding tissue [85,86,87]. Scaffold ECM molecules, pore sizes, and other properties have been demonstrated to affect cell survival, proliferation, adhesion, cellular uptake, protein adsorption, synthetic molecules (nanocrystal/fibers) accumulation, and sequestration in organelles, etc. [88]. Decellularized tissues, cellular genomic material, cellular protein, lipids, etc., have been labeled as immunogenic molecules [89,90]. The advantage of decellularized scaffolds over synthetic scaffolds is the fact that decellularization can reproduce, to a greater extent, an accurate structure of the extracellular matrix, in which cells reside in vivo, and usually still possess some of the molecular and mechanical signals which induce cells to grow and proliferate in a much more tissue-specific manner. Several decellularization efforts have been focused on obtaining a scaffold devoid of immunogenic molecules through several techniques [91,92]. In this work, the level of cytotoxicity of the decellularized scaffold was measured at different levels of decellularization, in order to achieve safer/biocompatible constructs for in vitro cell studies. The results indicate successful decellularization at the 36-h time point, which recorded the lowest percentage of cell death compared to all other time points of decellularization. Data are shown in Figure 11 and Figure 12. 

#### 2.11.1. Qualitative Analysis of Cytotoxicity of DACS on Chondrocytes

As shown in Figure 11B,C, the DACS scaffold had a cytotoxic effect on chondrocytes seeded on it at the 12-h and 3-h decellularization time points, respectively, as seeded cells could be observed to be rounded and dead after 24 h of culturing. Notwithstanding, the 36-h decellularized DACS produced viable cells after the same time point of culturing. Chondrocytes could be observed attached to the scaffold with extended pseudopodia, as characteristic of their morphologies (Figure 11A). Five days of culturing shows more cells thriving, attaching and proliferating on the 36-h decellularized DACS scaffold (Figure 11D). After 15 culturing days, as shown in Figure 11E, chondrocytes are seen completely invading, infiltrating, and colonizing the DACS scaffold. Figure 11F shows the monolayer plate control seeded chondrocytes healthily proliferating at day 3 of culturing. 

#### 2.11.2. Quantitative Analysis of Cytotoxicity of DACS on Chondrocytes 

The DACS scaffolds, after varying time points of decellularization, were analyzed for their cytotoxic effects, via the assessment of the level of inhibition of the reseeded chondrocytes. This would enable the determination of the time point of decellularization at which the scaffold could be safe to be used as an implant for a recipient, especially in clinical application. As shown in Figure 12, the level of inhibition (34.47%) at the 3-h decellularization process was significant (*p*-value < 0.05) in the DACS scaffolds compared to the 36-h decellularized treatment (which recorded 5.42% inhibition). However, between the 36-h decellularization group and that of the monolayer control, there were no observed significant differences (*p*-value < 0.001). The native cartilage controls registered 46.05% inhibition, the highest inhibitory effect in all the treatment groups. This low inhibition, recorded in the 36-h group, was indicative of cytocompatibility of the DACS scaffold to chondrocytes growth after 36 h of decellularization. Having successfully shown that decellularized scaffolds were devoid of cellular protein (Figure 5A,B), cellular lipids (Figure 7A,B) and cellular DNA (Figure 6A,B) at the 36-h time point of decellularization, this cytocompatibility study reaffirms those findings, since it showed insignificant inhibition, as compared to the monolayer controls.

Therefore, this inhibitory study corroborates the removal of cellular protein DNA and lipids, as shown in Figure 5, Figure 6 and Figure 7, respectively. The higher inhibitory values observed for the 3-h time points are presumably due to lower decellularization. At such lower decellularization time points, the presence of these molecules (cellular proteins, cellular lipids and DNA) was recorded, which indicates the reason for the cytotoxic reactions observed in all treatments at the 3- and 12-h time points, as shown in Figure 11B,C and Figure 12. The average 5.42% growth inhibition at the 36-h decellularization time point could be attributable to cell handling, such as trypsinization, centrifugation and pipetting during experimentation.

### 2.12. Analysis of Collagen Expression 

Col2A1 is an essential gene in chondrogenesis [93,94]. This gene encodes for one component of type II collagen, called the pro-alpha1 (II) chain. Type II collagen gives structure and strength to connective tissues [95,96,97]. As demonstrated in Figure 13A, normal tissue formation is consistent with Col2A1 interacting with other collagen molecules, e.g., Col1A2, Col9A1 and 2, Col11A1 and 2, and ADAMTS3 [95,96,97,98,99,100,101,102]. It is also known to interact with Col10A1 (also known as Col X) [103]; however, this interaction is limited and only pronounced during chondrocyte hypertrophy. In a cartilage disease condition, such as rheumatoid arthritis or osteoarthritis, there is a significant shift in interactivity, with Col2A1 interacting solely with Col10A1, without any interaction with Col1A2, Col9A1 and 2, and Col11A1 and 2, leading to poor cartilage formation and abnormal composition [104,105]. Furthermore, as presented in the degenerative cartilage, in the Col2A1 interaction network (Figure 13B), Col2A1 has a direct and higher frequency of interaction with one of the potent initiators of cartilage destruction—MMP13. MMP13 is known to be a positive indicator of osteoarthritis and other cartilage degeneration [106,107]; it has a role in the degradation of collagen in the matrix [108].

As shown in Figure 14A, Collagen 2A1, 5A1, 6A1 was upregulated in chondrocytes grown in the 3D DACS scaffold. Figure 14B shows the western blot analysis of collagen 2A1 protein, expressed by chondrocytes seeded on the DACS scaffold at various days of culturing. These molecules drive the chondrocyte remodeling of the scaffold’s ECM, ECM formation and deposition, and play a crucial role in chondrocyte adhesion. ECM secreted by chondrocytes seeded on the decellularized cartilage scaffold is known to contain higher levels of ACAN and Col2A1 (markers for aggrecans and type II collagen, respectively) [30], and this agrees with our findings, as higher expression levels of Col2A1 were observed in the DACS scaffolds than in the plate controls, as depicted in Figure 14A. While a normalized mean expression of 1.92 was observed for the control plate group, an approximately 3-fold increased expression was observed in the DACS-seeded chondrocytes, registering 5.61 mean normalized mRNA expressions. Probing for Col2A1 in DACS-seeded chondrocyte’s lysate, using an anti-Col2A1 antibody, we recorded significant band intensities for days 3, 5, and 15 of culture, with day 15 having the highest intensity (Figure 14B).

### 2.13. Higher Expression and Deposition of Col2A1 in DACS than Monolayer Control

As depicted in Figure 15, there were higher expressions of Col2A1 in DACS than the monolayer control, with increasing culturing days. The extracellular role of Col2A1 has been well-characterized [109,110]. Col2A1 accumulates in the endoplasmic reticulum post-translationally, before which, it is transported extracellularly [111]. 

## 3. Discussion

Most cartilage-related diseases result in drastic alteration within the ECM, which orchestrates disease progression or even the cause of the symptoms itself [112]. In this work, a xenographic scaffold, produced from avian articular cartilage, was reseeded with human chondrocytes, and the cells improved in viability and converted the scaffold into a neo-cartilage. The resulting data shows that the DACS pore sizes and distributions and ECM thickness in young and old avian articular cartilages vary significantly, which could have implications on cartilage degeneration. Scaffolds produced from older avian cartilage will have lesser porosity than the young cartilage scaffolds, since porosity is directly related to the pores created from dislodging the avian tissue cells. We infer from these results that reseeded cells’ migration, invasion, and infiltration of the scaffolds will be much faster in the young avian articular cartilage’s scaffolds than the old ones, since there were far more pores per unit area of the scaffold in the young avian cartilage than the old. However, such a comparative study was not carried out here. 

Most religious groups prohibit the use of porcine or bovine parts for human tissue grafts; however, the use of avian tissue has no recognized religious or moral restrictions, unlike porcine and bovine tissues [43]. Thus, avian decellularized cartilages can be repurposed for human cartilage regeneration. The ease of producing DACS, as well as the availability of avian cartilage, makes it a suitable alternative for engineering human cartilage tissue. The biophysical characteristics of the articular avian cartilage tissue used in this study adequately support human chondrocyte growth and neo-cartilage development. The decellularized construct is biodegradable, biocompatible and the production process has no biosafety concerns.

Furthermore, the avian cartilage tissue has a similar microstructure and collagen fiber arrangement to the human cartilage structure [57]. Our use of avian cartilage introduces an exciting possibility of repurposing these commonly available biological tissues, which are regarded as waste and discarded in slaughterhouses, for cartilage engineering and osteoarthritis and rheumatoid arthritis treatment. Collagen II and GAG are critical in guiding cellular survival, attachment, proliferation, differentiation, and migration and are significant components of the ECM of cartilage [48]. Hence, in cartilage regeneration, the intent is to have neo-cartilage, consisting of high collagen II and GAG contents, comparable to natural cartilage. While decellularized bovine cartilage has been repopulated with mesenchymal stem cells from mice (MSCs), and decellularized human cartilage also reseeded with MSCs in a similar study [59], this work is the first to reseed decellularized avian cartilage with human chondrocytes. The level of cytotoxicity of the decellularized scaffold at 36-h decellularization showed safer levels for in vitro cell studies and the possibility of biocompatibility in vivo, if employed. Furthermore, the high amount of Col2A1 deposition extracellularly indicates collagen deposition and the scaffold’s ECM remodeling by the reseeded chondrocytes. Hence, it might be safe to propose that xenographic avian articular cartilage, when decellularized, could be a suitable alternative for human cartilage tissue regeneration. 

This work, however, contains some drawbacks. For instance, chondrocytes’ morphology and alignment, with respect to the articular surface, is crucial for the tissue’s performance [47]. While the major interest of this work was to replace the avian chondrocytes with human chondrocytes, such that the physical and epigenetic cues provided by the DACS serve to induce chondrocytes towards cartilage phenotypes, such observation was not recorded. The reseeded cells not only attached to the ECM of the scaffold, but also infiltrated and broke up the ECM in elongated phenotypes. The scaffold’s pores did not serve the role of resident cavities for the cells but rather allowed cells to penetrate and infiltrate the scaffold. The usual phenotypes of chondrocytes are oval, rounded or polygonal, or flattened, depending on the exact location in the cartilage, but rarely elongated and fibroblastic in structure. Nonetheless, these elongated phenotypes were the predominant phenotypes observed in this study. Currently, it is unclear why reseeded cells exhibited this characteristic, instead of recapitulating the exact phenotypes of cartilage. Our laboratory has begun investigating new methods to induce reseeded chondrocytes on DACS to achieve oval/round shape phenotypes. 

One of the major cartilage regeneration procedures is the Autologous Matrix-induced Chondrogenesis (AMIC^®^), for which microfracturing of the subchondral bone is made to recruit blood into a biodegradable natural collagen type I/III membrane Chondro-Gide^®^ (Geistlich Pharma AG, Wolhusen, Switzerland) to host and hold the superclot generated by the microfracturing process [113]. A study by Volz et al., 2016, demonstrated that Chondro-Gide^®^, when sutured to the defect site, was effective in cartilage regeneration [113]. However, since Chondro-Gide possessed no chondrocytes and depended on the recruitment of mesenchymal cells from the subchondral bone to differentiate into cartilage tissue, it is both time consuming (2 to 5 years) and sometimes ineffective. 

In another study, Kim et al., 2021, treated osteochondral defects by synovium-derived mesenchymal stem cells (SMSCs), encapsulated in a HA/collagen/fibrinogen composite scaffold. Their data proved scaffold-supported articular cartilage regeneration and upregulated integrin β1 and actin remodeling [114]. A similar study demonstrated that a scaffold fabricated from human dermal-derived collagen showed positive cartilage regeneration potential [115]. In these studies, however, scaffolds have been seeded with mesenchymal stem cells, which do not thoroughly differentiate into chondrocytes, thereby affecting the quality of the neocartilage formed. In this work, we are the first to culture human chondrocytes in a decellularized avian cartilage, and in an in vitro condition, we observed the formation of a neocartilage from the culture. Our study suggests that within two weeks of recellularization of DACS, neocartilage was obtainable, and so could be used for transplants to patients, thus, drastically reducing the wound healing time. Secondly, since scaffolds are directly seeded with chondrocytes instead of stem cells, as demonstrated by other researchers, this cut off the need for special treatment for chondrogenic induction. 

A significant weakness of this study is the absence of an in vivo study using DACS, to show and compare with existing methods and scaffolds in the market. Secondly, an in-depth analysis of other regulatory factors, apart from collagen, associated with proper cartilage regeneration, needs to be assessed. Based on the availability and ease of producing DACS, it is appealing to speculate that DACS might function as superior scaffolds to existing scaffolds in the market, in their ability to support and induce cartilage regeneration via chondrocyte-based cartilage tissue regeneration, since these scaffolds have been demonstrated in this work to encourage chondrocyte attachment, proliferation and induced ECM collagen production. 

In summary, the decellularized avian cartilage constructs provide an excellent and functional cartilage construct for transplantation purposes in human cartilage regeneration. Furthermore, this construct could be employed in ex vivo experimentations, since it recapitulates the microstructure of the cartilage ECM. The next step forward is to study the effect of transplanting the recellularized scaffolds into an animal model, in order to investigate the immunological and physiological events characterizing the tissue’s healing and development.

## 4. Materials and Methods

### 4.1. Decellularization of Avian Articular Cartilage as Scaffold (DACS)

#### 4.1.1. Cryosectioning of Decellularized Avian Articular Cartilage

The cartilage is a thick and compart tissue, thus decellularizing this tissue is time consuming and most times less effective, however, to enhance the decellularization process, thin sections of the native cartilage tissue were made using cryosectioning. Prior to sectioning, the samples of articular cartilages from young avian (<30 days) and old avian (>30 days) were extracted from slaughtered chickens (Gallus Gallus domesticus) obtained from the local abattoir. The sectioning cartilage specimens were mounted on a cryostat chuck using Tissue Freezing Medium (Leica Microsystems, Deutschland, Germany) and sectioned into 500 µm thin specimens using the HM315 Rotary Microtome (Microm, Walldorf, Germany). Cryosections were used for the decellularization process. 

#### 4.1.2. Decellularization of Sectioned Cartilages

The decellularization process was employed to reduce the immunogenic components of the cartilage tissue by eliminating the cellular elements such as proteins, nucleic acid, and lipids while maintaining the mechanical and bioactive properties of the cartilage ECM. Avian articular cartilage tissues were extracted from both young (<30 days old) and old (>30 days). The thin sections of avian articular cartilages with dimensions of approximately 2 mm by 5 mm by 0.5 mm for width, length and thickness respectively were used for the decellularization process.

In this decellularization process, a comparative study was done by decellularizing the thin sectioned cartilages using Sodium Dodecyl Sulfate (SDS) (Fisher Scientific, Fair Lawn, NJ, USA) and then Triton-X. Decellularization processes were carried out on a rocking platform (Variable Rocker, MIDSCI, Benchmark Scientific Inc., Edison, NJ, USA) under maximum rocking at room temperature. First, the microtome-sectioned cartilage tissue was placed in tissue packets and immersed into the 1% SDS or 1% Triton-X solution for the set time points of 3, 12, 24, 36, and 48 h. Following detergent treatment, the cartilage specimens were subjected to 5 cycles of washing for 10 min each with distilled water and then followed by sterilization in 70% ethanol. Lastly, sterile specimens were dialyzed against dH_2_O (5 changes of dH_2_O for 20 min each) to remove residual ethanol.

#### 4.1.3. Scanning Electron Microscopy (SEM) 

The preparation of the cartilage for the SEM was carried out following a similar procedure described by Li et al., 2021 [116] with few modifications. In brief, the decellularized sections of the cartilage specimens were fixed with 10% formaldehyde solution at room temperature for 10 min, then washed with 1X PBS solution three times, and followed with serial dehydration in 50%, 70%, and 95% absolute ethanol solutions for 10 min each. Then specimens were dried and sprayed with gold using EMS Quorum (EMS 150R ES, Quorum Technologies Ltd., Laughton, UK) ion-sputtering instrument and observed through Analytical Scanning Electron Microscope (SEM) (JEOL JSM-6010LA, Tokyo, Japan) installed with IntouchScope software (JSM-IT200 InTouchScope™ SEM Series, JEOL Solutions, Tokyo, Japan).

### 4.2. Lipid Extraction and Quantification 

To extract lipids from the decellularized and native cartilage tissues, we followed Folch’s extraction procedure as described by Castro et al., 2013 [117] with few modifications. That is, 0.1 g of the sample (DACS or native cartilage tissue) in a 15 mL tube was homogenized using Branson Sonifier (FlackTek, Inc., Landrum, SC, USA) for 2 min on ice. Next, 1.2 mL of a 2:1 chloroform/ methanol (2:1, *v*/*v*) solution was added and vortexed for 30 s. This was followed by the addition of 0.4 mL of methanol and vortexed for another 30 s. The mixture was filtered through Whatman filter paper. The filtrate was transferred to a fresh 15 mL tube. Then 0.8 mL of chloroform and 1.90 mL of NaCl (0.73%) solutions were added and vortexed for another 30 s and the mixture was allowed to phase separate at room temperature under the flow hood. The organic phase and the lipid phase were collected separately. This was followed by sitting the tube containing the lipid phase on a water bath set at 40 °C to evaporate for 3 h until only the lipid remained. The weight of lipid was then determined. Lipid content was calculated based on wet weights of samples. A control extraction with no tissue in the tube was run and the values used to normalize the experimental results.

### 4.3. Sulfated GAGs Extraction and Quantification 

To extract and quantify sulfated GAGs, the procedure used by Le et al., 2017 [118], Burkhardt et al., 2001 [119] were employed with few modifications. For the sulfated GAGs quantitation, 4 samples each from the extracted cartilage of the young that weighed approximately 0.86 g and the old which weighed approximately 1.11g were decellularized. Four cartilages served as native cartilage controls. Both decellularized and native cartilages were analyzed for sulfated GAGs content by using the following procedure: 1 mg of thin sections of the native or the decellularized cartilages were homogenized using Branson Sonifier 250 (Branson Ultrasonics, Shanghai, China) and dried. The dried samples were placed in a 6-well plate and digested with papain overnight at 60 °C in 1% SDS, 50 mm Tris/HCl, pH 8 buffer on a rocker. The digested samples were heat-inactivated at 90 °C (Edvotex water bath, Bethesda, MD, USA) for 10 min and NaCl added to a final concentration of 0.1 M and centrifuged at 12,000 rpm for 10 min. The mixture was filtered through Whatman filter paper. The solution was dialyzed by cellophane membrane with size of 14kDa for 3 h. Then, finally, the filtrate was allowed to precipitate overnight for GAGs at 5 °C by the addition of ethanol 99% (*v*/*v*) and the sulfated GAGs dried at 60 °C. The resulting powder was weighed, and the data were presented as ± Standard deviation.

### 4.4. Seeding of Chondrocytes on DACS Scaffold

The decellularized avian articular cartilage scaffolds (2 mm by 5 mm by 0.5 mm dimensions of width, length and thickness respectively) after sterilization and dialysis using distilled water were aseptically placed into the wells (2 scaffolds per well) of 24-well tissue culture plate (TPP Techno Plastic Products, Trasadingen, Switzerland), and human chondrocytes cells (HC Cat # 402-05f, Cell applications, INC) at 1 × 10^4^ per well in chondrocytes growth medium (Cat# 411-500, Cell, application, INC) were seeded on the scaffolds. Determination of the number of cells per well during seeding was carried out using the Countess II cell counter instrument (Life Technologies Corporation, Bothell, WA, USA). The cells were then incubated in Forma Series 3 Water Jacketed CO_2_ Incubator (ThermoFisher Scientific Inc., Marietta, OH, USA) for 24 h, 48 h, 72 h, 120 h, 240 h, or 360 h followed by cell harvest for downstream experimentation. Controls consisted of all the above procedures except the absence of scaffolds in them. 

### 4.5. Protein and Nucleic Acid Assessment of DACS

#### 4.5.1. Protein Assessment of DACS

For this part of the process, 5 mg each from native cartilage and DACS were frozen in lysis buffer (50 mM Tris-Cl, 1 mM PMSF, 1 mM DTT) and then subjected to homogenization using the Branson Sonifier 250 (Branson Ultrasonics, Shanghai, China) on ice. Sonification was done at output maxima of 5 and for 3 min at constant duty cycle. The lysate was loaded into 10% SDS polyacrylamide gel and run at 100 volts in Bio-Rad Mini-Protean 3-Cell electrophoresis chamber for 50 min. Gel bands were visualized by Coommasie blue staining (Life Technologies Corporation, Carlsbad, CA, USA) and gel images acquired by Chemi-Doc XRS (Bio-Rad Laboratories, Hercules, CA, USA) loaded with Quantity One software (Quantity One version 4.6.9, Bio-Rad Laboratories, Inc., Hercules, CA, USA). Protein bands were quantified using Image J (ImageJ 1.53, Wayne Rasband and Contributors, National Institute of Health, Bethesda, MD, USA), and values expressed as mean densitometric values ± standard deviation.

#### 4.5.2. Nucleic Acid Assessment of DACS

The genomic DNA (gDNA) of both native and decellularized avian cartilage was extracted from 5 mg sample each and in 3 replicates (*n* = 3) and employing the DNeasy tissue kit (Cat# 69504, Qiagen Sciences, Germantown, MD, USA), the tissue DNA of the native and decellularized specimens were extracted. After DNA extraction, DNA was quickly stored at −20 °C until further application. Quantitation of DNA was achieved via 1% agarose gel analysis using agarose low EEO electrophoresis grade (lot# 976260, Fisher Biotech, Fairlawn, NJ, USA) heat dissolved in 1X TAE buffer prepared from 50X TAE stock (Cat# 166-0742, Bio-Rad, Hercules, CA, USA) and run at 90 volts for 60 min using the Bio-Rad Mini-Sub Cell GT electrophoresis chamber (Bio-Rad Laboratories, Hercules, CA, USA). Bands were visualized using 1% ethidium bromide staining for 20 min and dH_2_O wash, and gel images acquired by Chemi-Doc XRS (Bio-Rad Laboratories, Hercules, CA, USA) installed with Quantity One software (Quantity One version 4.6.9, Bio-Rad Laboratories, Inc., Hercules, CA, USA). Quantitative measure of bands’ intensities were achieved by Image J, and densitometric values plotted into bar charts. Data were averaged and expressed as the mean densitometric values of DNA bands ± Standard deviation. 

### 4.6. Histochemistry

Samples from the microtome-sectioned native cartilage (*n* = 10) and microtome-sectioned and decellularized DACS (*n* = 10) were stained with Alcian Blue, Oil Red, and Flash Blue to detect GAGs, oil droplets, and tissue’s nucleic acid. In addition, several stained sections were microphotographed using TCM 400 (Labo America, Inc., Fremont, CA, USA) or EVOS FLC microscopes (Life Technologies, Carlsbad, CA, USA).

### 4.7. In Vitro Assay of DACS Biocompatibility via the Use of Alamar Blue for Cell Inhibition

Alamar Blue is a reagent which contains the cell permeable, non-toxic, and fluorescent blue dye called resazurin. Alamar Blue is a useful non-toxic dye used for cytokine bioassays, in vitro cytotoxicity, and cell viability assays (3-(4,5-dimethylthiazol-2-yl)-2,5-diphenyl-tetrazolium bromide) [120]. Alamar Blue monitors the reducing environment of the living cell [121]. The active ingredient is resazurin (IUPAC name: 7-hydroxy-10-oxidophenoxazin-10-ium-3-one) and is water-soluble and permeable through cell membranes. Hence, an uninterrupted monitoring of cells in culture can be carried out. This assay was employed to assess the toxicity of the native and decellularized scaffolds via measuring the inhibition of chondrocytes viabilities in them. The Alamar Blue assay was employed for the study according to the manufacturer’s protocol. For inhibitory analysis, chondrocytes at the density of 10^4^ were seeded into wells of 24-tissue plates containing native cartilage, 3-h decellularized scaffold, 36-h decellularized scaffold and a monolayer which served as control. Plates containing cells were incubated overnight. Afterwards, cells were incubated with the Alamar blue reagent for 6 h. The plates were then read at an absorbance of 570 nm and 600 nm. Similar assay has been employed in the assessment of cell viability and toxicities [122]. Cells in DMEM media without scaffolds and cells were read, averaged, and used for the normalization of the data. Calculations for inhibition were carried out for five replicate experiments. Data reported as ± Standard deviation. 

Also, microphotographs were taken daily from each experimental group to monitor cell survival and progression in each experimental group or the control.

### 4.8. Alkaline Phosphatase (ALP) Assay for Chondrocytic Proliferation

ALP was used to assess cell proliferation, and this was assessed by the determination of alkaline phosphatase conversion of the substrate. To evaluate ALP activity, chondrocytes cultured in 96-well plates were washed thrice with (PBS, pH 7.4), and 0.2% SDS was added. The Enzyme activity was determined by measuring the cleavage of 10 mmol/L p-nitrophenyl phosphate (pNPP) in 1 mol/L diethanolamine buffer (pH 9.8) and 0.2 mmol/L MgCl_2_ at 37 °C. After incubation of the mixtures at 37 °C for 30 min, 100 μL of 0.1 M NaOH was added to stop the reaction. Similar assays have been used by other researchers [123,124]. The ALP determination using an optimized substrate concentration and 2-amino-2-methyl-1-propanol as the buffer is a colorimetric assay. In magnesium and zinc ions, p-nitrophenyl phosphate is cleaved by phosphatases into phosphate and p-nitrophenol. The p-nitrophenol is released in direct correlation to the catalytic ALP activity, and this kinetics were assayed by measuring the increase in absorbance at 409 nm. Thus, this assay can be expressed as;

(4 − Nitrophenyl phosphate (colorless) + ALP) = (4 − Nitrophenol (yellow color))

Experiments were performed in triplicate.

### 4.9. Gene Expression Analysis

The total RNA from each time point of each treatment group of the cultured chondrocytes (*n* = 3) was extracted using the Rneasy kit (Qiagen Sciences, Germantown, MD, USA). Then, RNA was dissolved in RNase-free water, and the concentration was determined by Nanodrop (1000 spectrophotometer, ThermoFisher Scientific, Madison, WI, USA). Then, approximately 100 ng of RNA from each sample was then reverse transcribed into cDNA using SSIV Cell Direct cDNA Synthesis kit (lot# 00916637, Invitrogen, ThermoFisher Scientific, Vilnius, Lithuania), followed by expression analysis for specific genes. The following genes were analyzed: Col2A1, Col4A2, Col5A1, and Col6A1 using CFX96^TM^ real-time PCR (Bio-Rad Laboratories, Hercules, CA, USA) as described by (Di Meglio et al., 2010). Three samples from each set experimental group were tested using Glyceraldehyde-3-phosphate dehydrogenase (GAPDH) gene (Eurofins genomics, Louisville, KY, USA) as a housekeeping gene control. Real-time QPCR analysis of collagen expression of reverse-transcribed cDNA was carried out using the primers listed in Table 1 and powerup™ SYBR™ Green Master Mix (lot# 00914521). All primers used for this analysis are listed in Table 1. The QPCR process was carried out using Bio-Rad CFX96 Real-Time PCR Detection System (Bio-Rad Laboratories, Hercules, CA, USA). A similar method has been employed in the analysis of genes expressed in other scaffolds [125,126]. Comparisons of target gene expression in the samples were carried out using the cycle threshold (Ct) normalized to the housekeeping gene GAPDH.

### 4.10. Immunofluorescence Assay of Chondrocytes Seeded on Scaffolds

Col2A1 is a significant ECM formation and remodeling molecule [127,128]. Therefore, assessing this molecule’s synthesis, deposition, and localization is an essential measure of ECM remodeling [129]. Cells on scaffolds or control plates were fixed, permeabilized, and incubated with Col2A1 rabbit primary antibody (catalog# PAB19147, Abnova, Taipei, Taiwan). All reagents were purchased from Thermo Scientific unless otherwise noted and by following published protocols from manufacturers and published works [129,130]. The expression of Col2A1 molecules was analyzed and visualized using this microscopic method. To carry out this process, samples (reseeded DACS or monolayer plate control) were formalin-fixed and permeabilized for 3 h with a permeabilizing solution containing 1% BSA and 0.1% Triton-X in PBS, then blocked with 1% BSA and incubated overnight with the primary antibodies. After incubation, samples were washed thrice with PBST (10 min per wash) and incubated in HRP-conjugated secondary antibodies against rabbit for 1.5 h. The secondary antibody used was goat anti-rabbit, HRP secondary (lot# K3491186, Millipore Sigma). Cells were visualized using Quanta Red™ Enhanced Chemi-fluorescent HRP substrate (lot# VG306301, ThermoFisher Scientific, Waltham, MA, USA). Phalloidin 488 (lot# 2129460, Invitrogen, Thermos Fisher Scientific, Waltham, MA, USA) staining was utilized in detecting F-actin, while DAPI (4′6-diamidino-2phenylindole) (lot# 2116140, Invitrogen, Thermos Fisher Scientific, Waltham, MA, USA) was used for nuclear staining. Immunofluorescent analysis via microscopy was carried out using the EVOS FLC microscope (Life Technologies, Carlsbad, CA, USA).

### 4.11. SDS-PAGE and Western Blotting

Cell lysates were made from DACS-seeded chondrocytes and also from the monolayer plate controls, and lysates were electrophoresed using 10% SDS-polyacrylamide gels. Blotting was done using the transfer buffer consisting of 50 mM Tris, 40 mM glycine, 20% methanol, 2% SDS, and transferred to a PVDF membrane using a Bio-Rad transfer cell. After transfer, PVDF membranes were washed briefly in TBST buffer consisting of 150 mM NaCl, 50 mM Tris-HCl, pH 7.4, 0.1% Tween 20, followed by blocking for 2 h in 5% BSA. Anti-Col2A1 (catalog# PAB19147, Abnova, Taipei, Taiwan) was diluted 1:1000 in 5% BSA in TBST and incubated overnight at 4 °C. Membranes were washed thrice and then counter incubated using horseradish peroxidase (HRP)-conjugated (goat anti-rabbit IgG (lot# K3491186, Millipore Sigma, St. Louis, MO, USA)) secondary antibody for 1 h, and HRP signals were visualized by chemi-luminescence using Chemi-Doc XRS installed with Quantity One Software (Quantity One version 4.6.9, Bio-Rad Laboratories, Inc., Hercules, CA, USA).

### 4.12. Image Acquisition and Processing

The immunofluorescent images of para-formaldehyde-fixed cells stained with DAPI were acquired by EVOS FLC microscope (at 50 μm, Zeiss 510, 100×/1.3 oil objective) (Life Technologies, Carlsbad, CA, USA), whereas phase contrast microphotographs were acquired using TCM 400 inverted Microscope (Labo America, Inc., Fremont, CA, USA).

#### Pore Size Analysis

Thin sections of DACS constructs were imaged at 50 μm magnification using the high-resolution objective lens of EVOS FLC microscope (Life Technologies, Carlsbad, CA, USA). Images were then transferred to image J and then processed for pore sizes, ECM thickness and the 3D volumetric reconstruction of the imaged DACS. 

### 4.13. Biomechanical Behavior Testing

Biomechanical behavior was studied using DMA 850 dynamic mechanical analyzer (TA Instruments, New Castle, DE, USA) equipped with a shear sandwich fixture. The shear plate was placed between rectangular samples with a length of 5 mm, a width of 5 mm, and a thickness of 1 mm. A stress ramp was performed by increasing the stress from 0 to 10 N at a rate of 1 N/min. The linear region of the stress vs. strain curve was used to calculate Young’s modulus. 

### 4.14. Statistics

All statistical data except indicated are expressed as mean ± Standard deviation. Statistical analyses were performed using OriginPro Plus version 2021b (OriginLab Corporation, Northampton, MA, USA) or Microsoft Excel 2013 (Microsoft, Redmond, WA, USA). Statistical Analysis was performed using paired *t*-test. Acquired images were analyzed and image qualities readjusted by image J.

## 5. Conclusions

We decellularized avian articular cartilages and employed them as novel biological scaffolds for human articular cartilage engineering in vitro. The decellularized scaffold achieved the main requirements of an effectively decellularized tissue, having undetectable nucleic acid, cellular protein, lipids content. We also demonstrated, via immunohistochemical images, as well as by scanning electron microscopic images, that these scaffolds possess key features, such as a high degree of porosity and interconnectivity, thus, allowing for reseeded chondrocytes to attach to, infiltrate and colonize the scaffolds. The assessment of the sGAGs content of the DACS revealed a relatively high amount of sGAGs, which are crucial for the performance of the scaffolds’ mechanical properties (that were demonstrated to be comparable to the native tissue), since their presence in these decellularized scaffolds lends support to the ECM’s integrity of the scaffold, thus, allowing cells to attach and interact with the scaffold. The compelling evidence for the high efficiency of these scaffolds at human cartilage regeneration is the relatively short time it took for chondrocytes to infiltrate and fully colonize the scaffolds to form neo-cartilage. This high efficiency of the DACS in neo-cartilage formation is an indication of possible superior performance of these scaffolds compared to available cartilage regeneration procedures in the market, which take far longer time (for instance, it takes 2 to 4 years for Chondro-Gide in an in vivo condition to heal wounded cartilage), thus, addressing the major hurdle that affects most synthetic biomaterials, as well as the microfracture procedures employed in cartilage regeneration. Additionally, DACS are biological materials and, therefore, are fully biodegradable and bio-resorbable. These scaffolds reduce/eliminate bioethical concerns when used as in vivo models, and the raw material for their production is commonly available, allowing for low cost production; they are easily constructed within a short time. These scaffolds recapitulate the 3D tissue complexity in an in vivo condition and they can be tunable via the addition of factors (molecular/chemical cues) that can be integrated into DACS to create desirable scaffolds of interest. The DACS have been shown to have eliminated molecular factors, antigens (including parasites, bacteria and viruses) that could potentiate tissue rejection during transplantation of constructs or cause zoonotic transfer of infectious agents to recipients after implantation. 

Our analysis of collagen biosynthesis, via both QPCR and western blotting, revealed that DACS upregulated the synthesis of these crucial ECM remodeling molecules, thus, ensuring that chondrocytes quickly reformed the cartilage tissue of desirable features compared to the native tissue, eventually replacing the DACS with newly synthesized cartilage ECM. The accrued data of the DACS study support the hypothesis that decellularized avian cartilage scaffolds, despite originating from different species, could potentially provide the appropriate microenvironment to enable human chondrocytes to grow, proliferate and form proper human cartilage for cartilage transplantation. Nonetheless, this work must be further proven in an in vivo model to ensure its efficiency and effectiveness as a cartilage regenerative tool. Currently, DACS holds great potential utility, in both clinical and academic settings, additively supporting the new resurgence in decellularization in general, as a groundbreaking technology in tissue and organ regenerative efforts.

In conclusion, we used SDS to generate decellularized avian cartilage scaffolds, which showed excellent biocompatibility with human chondrocytes in vitro. The avian cartilage tissue microstructure and ECM composition were not adversely altered, and the sGAG composition of the constructs was well preserved. The recellularized scaffolds showed excellent ECM molecule synthesis and high deposition of collagen, with increasing culturing days. The recellularized constructs, therefore, may provide excellent alternative grafts for transplantation in cartilage regeneration, to treat osteoarthritis or in cartilage injury.

## Figures and Tables

**Figure 1 materials-15-01974-f001:**
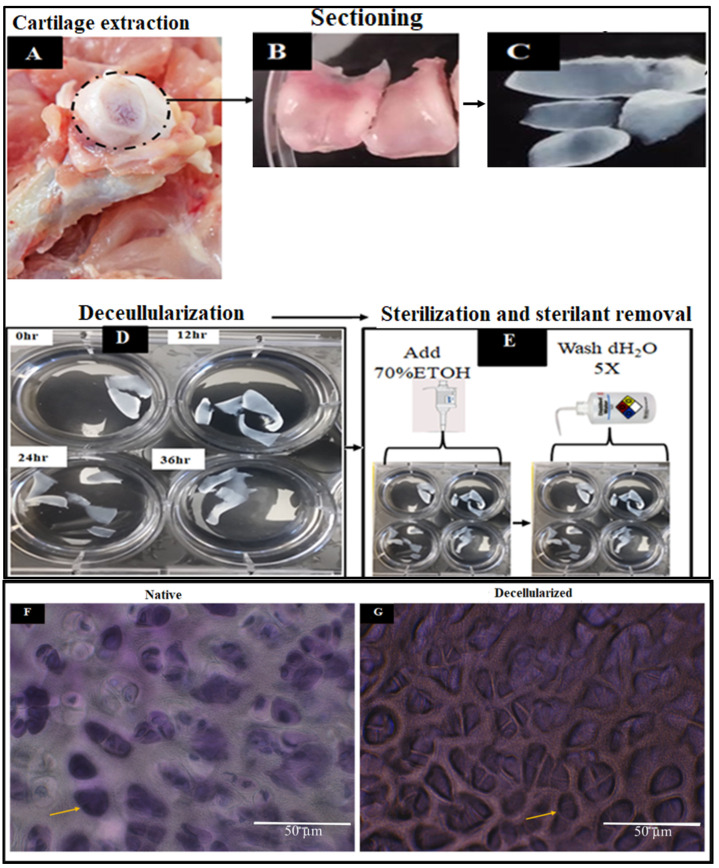
Panels showing the extraction, decellularization, and sterilization process of avian articular cartilage. (**A**) Avian articular cartilage at the knee joint. (**B**) Avian cartilage extracted and photographed. (**C**) Sectioned cartilage before decellularization. (**D**) Decellularization of sectioned samples. (**E**) Sterilization of sectioned and decellularized samples. (**F**) Flash Blue staining of sectioned cartilage revealing the cells’ nuclei in dark blue colors. (**G**) Flash Blue staining of decellularized cartilage revealed the absence of tissue cells/nuclei, showing empty pores.

**Figure 2 materials-15-01974-f002:**
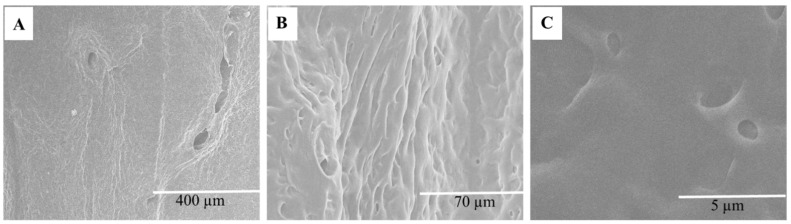
Panels showing the SEM images of decellularized avian articular cartilage. (**A**) Scale bar = 400 µm. (**B**) Scale bar = 70 µm. (**C**) Scale bar = 5 µm.

**Figure 3 materials-15-01974-f003:**
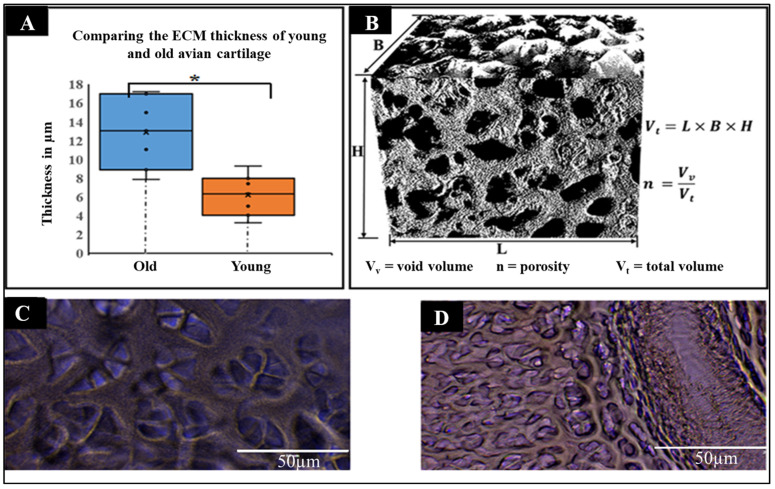
Dimensional analysis of Scaffold’s ECM and pores. (**A**) Quantitative measurements of ECM thickness of young and old avian articular decellularized cartilage. (**B**) 3D rendition of the decellularized scaffold, depicting the pore distribution and nano-grooves found on the ECM. (**C**) Stained scaffold image showing the pores of decellularized old avian articular cartilage. (**D**) Stained scaffold image showing the pores of decellularized young avian articular cartilage. Data are shown as mean ± SEM from 16 independent avian articular cartilages (8 young avian articular cartilages and 8 old avian articular cartilages. * *p*-value = 0.007. Scale Bar = 50 µm.

**Figure 4 materials-15-01974-f004:**
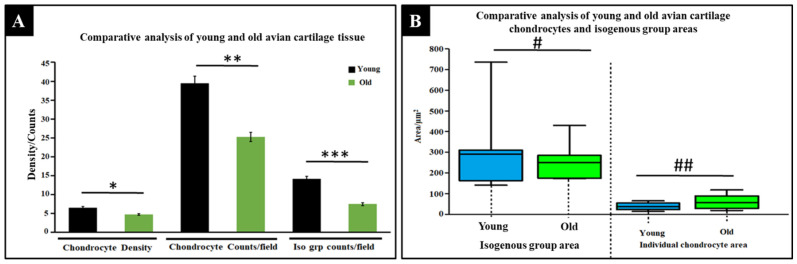
Comparative analysis of chondrocyte densities isogenous group areas and individual chondrocyte areas in young and old decellularized avian articular cartilages. (**A**) Cell densities of cartilages between young and old avian articular cartilages, chondrocyte’s count in the cartilage tissue of young and old avian articular cartilages, the number of isogenous groups found in young and old avian articular cartilages. (**B**) Assessments of densities of isogenous group areas and individual chondrocytes’ area of avian cartilage tissue. (**A**) *n* = 8, (**B**) *n* = 30. * *p* = *n* = 8, * *p*-value = 2.04824 × 10^−4^, ** *p*-value = 0.04152, *** *p*-value = 0.03128, # *p*-value = 0.93348, ## *p*-value = 0.50867. Legend: Iso group = Isogenous group.

**Figure 5 materials-15-01974-f005:**
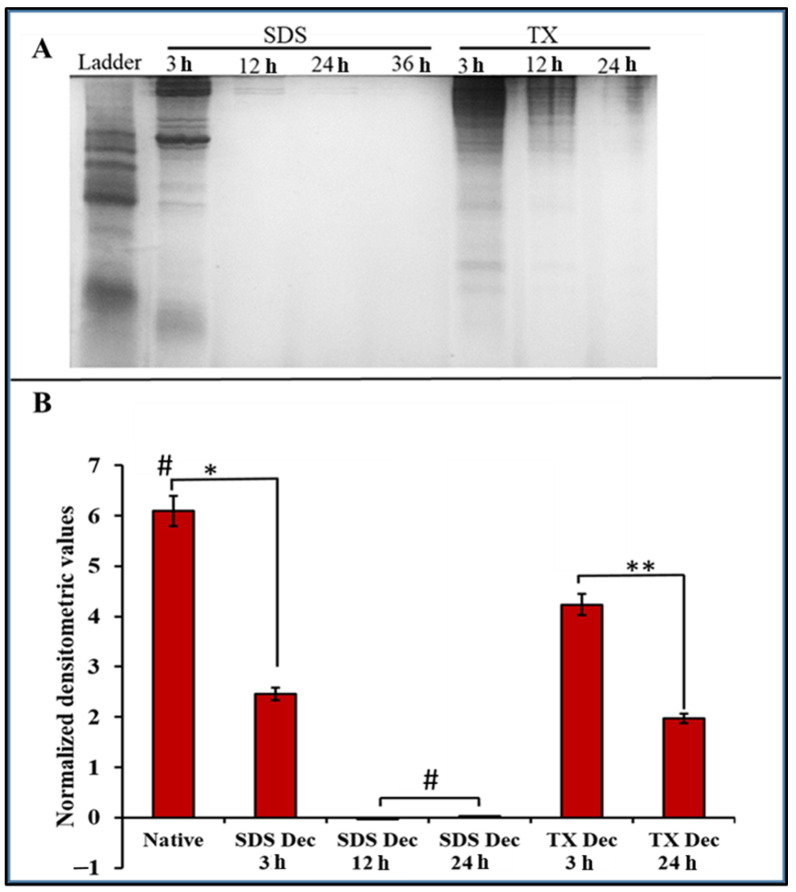
Charts showing the total cellular protein and nucleic acid content of decellularized scaffolds at different time points of decellularization. (**A**) SDS-PAGE of cellular proteins after decellularization in 1% SDS decellularization treatment or 1% Triton-X decellularization treatment. (**B**) Quantitative analysis of total cellular protein of decellularized cartilage tissue after 1% SDS decellularization treatment or 1% Triton-X decellularization treatment. * *p*-value = 0.0003, # *p*-value < 0.00007, ** *p*-value = 0.02742; *n* = 3.

**Figure 6 materials-15-01974-f006:**
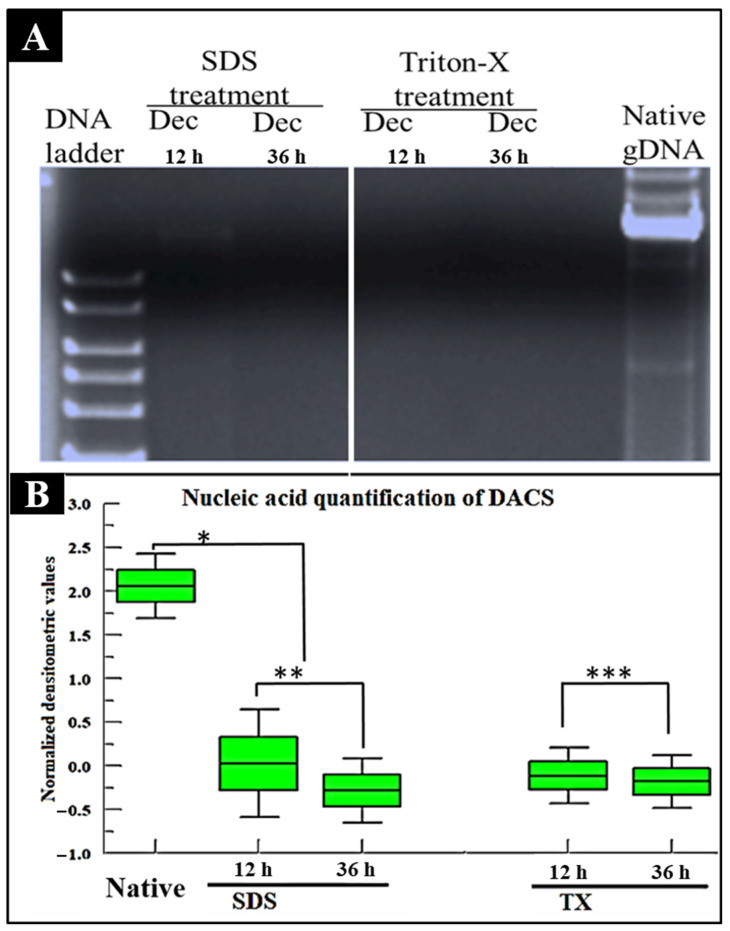
(**A**) Agarose gel analysis of DNA content of decellularized cartilage tissue after 1% SDS or 1% Triton-X treatment. (**B**) Nucleic acid quantitation of decellularized cartilage tissue for set time points after 1% SDS decellularization treatment or 1% Triton-X decellularization treatment. DACS = Decellularized avian articular Cartilage Scaffold. *p* < 0.05, *n* = 3. Data reported as mean values ± SEM, Student *t*-test. (*n* = 3, * *p*-value < 0.001, ** *p*-value = 0.631, *** *p*-value = 0.9408).

**Figure 7 materials-15-01974-f007:**
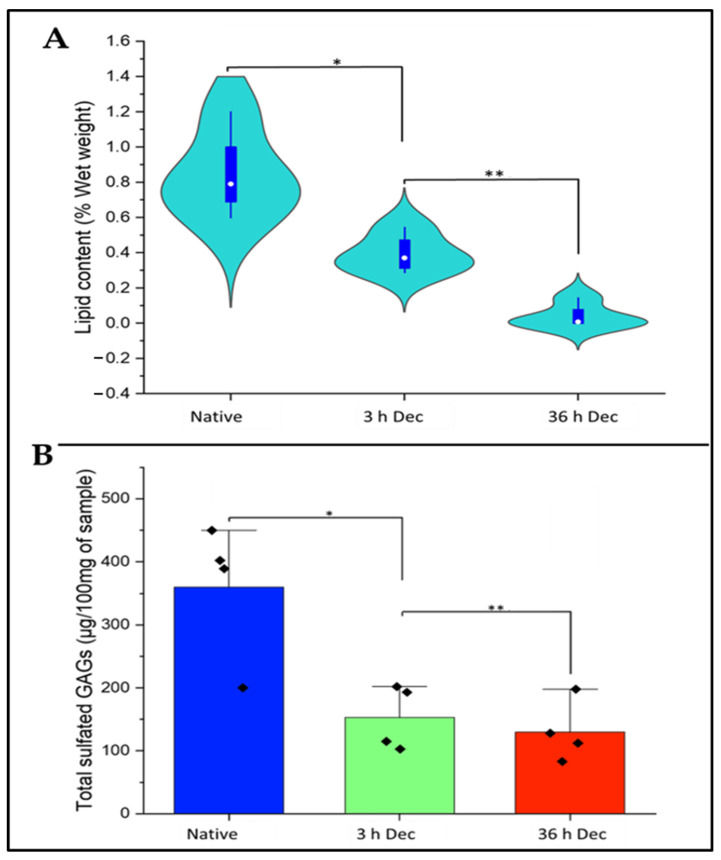
(**A**) Quantitative analysis of lipids in decellularized cartilage tissue. (*n* = 4, * *p*-value = 0.04864, ** *p*-value = 0.00283). (**B**) Quantitative analysis of sulfate GAG (sGAG) in decellularized cartilage tissue. (**A**) Data are shown as mean ± SEM. (*n* = 4, * *p*-value = 0.02458, ** *p*-value = 0.5408). (**C**) Comparative histochemical analysis of proteoglycan and tissue lipid content of decellularized avian articular cartilage. Alcian blue staining (AB). Oil red staining (OR).

**Figure 8 materials-15-01974-f008:**
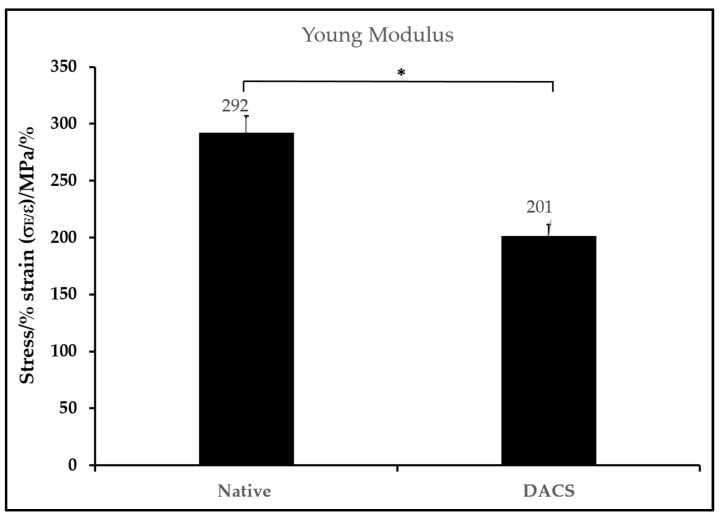
Mean Young modulus of DACS and the Native avian cartilage. Values are reported as mean ± Standard deviation, * *p* = 0.05, *n* = 3.

**Figure 9 materials-15-01974-f009:**
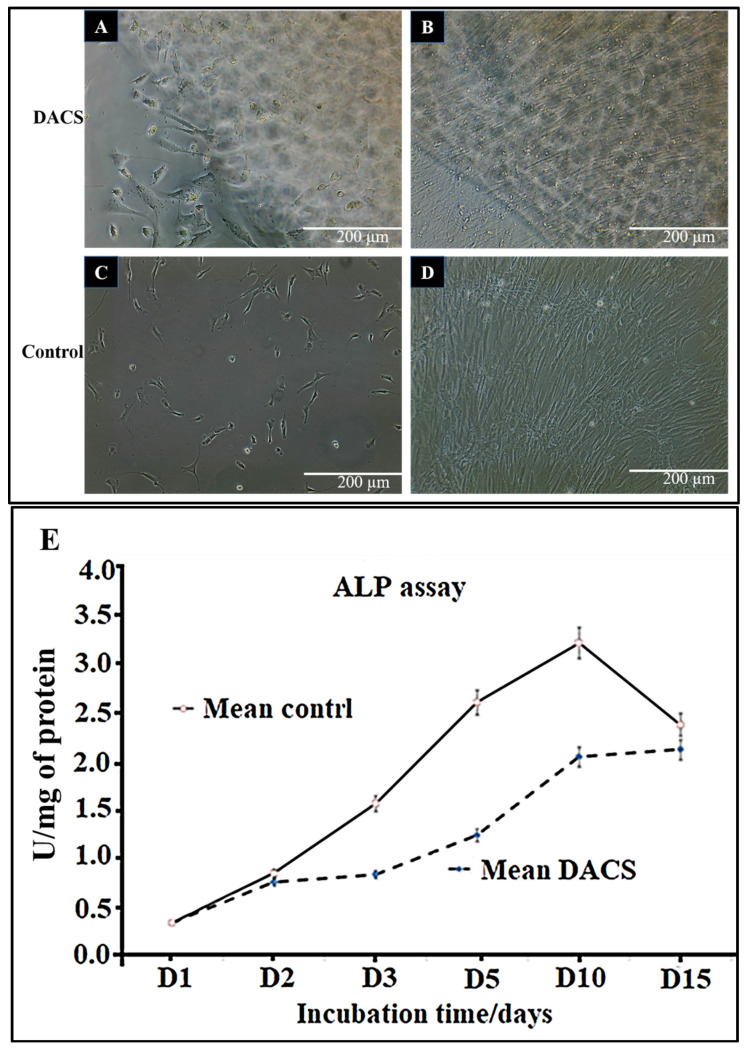
(**A**–**D**) Chondrocytes attachment and viability in scaffolds. (**A**,**B**) Chondrocytes on DACS at days 2 and 15 respectively. (**C**,**D**) Chondrocytes on monolayer plate control days 2 and 15 respectively. (**E**) Alkaline Phosphatase assay for chondrocytes proliferation for both control and DACS-seeded cells for varying culturing days. Data reported as mean ± Standard deviation.

**Figure 10 materials-15-01974-f010:**
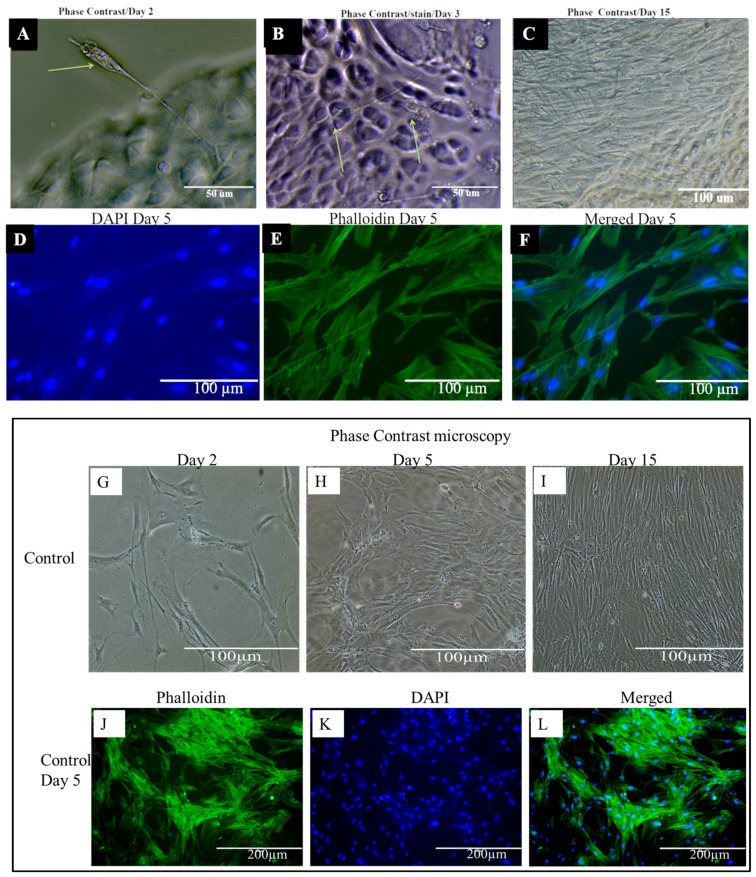
Morphological characteristics of chondrocytes seeded on DACS construct and plate Control. (**A**) Cells photographed after three days of culturing. Scale bar = 50 µm, 100 µm, or 400 µm as indicated on the micrograph. (**B**) Microphotograph of spear-shaped chondrocyte piercing through the DACS taken 50 µm resolution. Chondrocytes body on the plate, while the long pseudopodia stretched into the DACS. The six-well-plated controls did not show a similar phenotypic appearance; they were flat, with a large cell body and an extensive number of pseudopodia. (**C**) Complete colonization of scaffold by chondrocytes shown in phase contrast microphotographs day 5. (**D**) Colonization of scaffold by chondrocytes shown in blue via DAPI staining at day 5. (**E**) Colonization of scaffold by chondrocytes shown in green by phalloidin staining at day 5. (**F**) Shows the merging of the DAPI and phalloidin. (**G**) Phase contrast micrograph of chondrocytes in monolayer plate control, day 2. (**H**) Phase contrast micrograph of chondrocytes in monolayer plate control, day 5. (**I**) Phase contrast micrograph of chondrocytes in monolayer plate control, day 15. (**J**) Immunofluorescent micrograph of chondrocytes stained with Phalloidin in monolayer plate control, day 5. (**K**) Immunofluorescent micrograph of chondrocytes stained with DAPI in monolayer plate control, day 5. (**L**) Immunofluorescent micrographs of chondrocytes in monolayer plate control, day 5, merged.

**Figure 11 materials-15-01974-f011:**
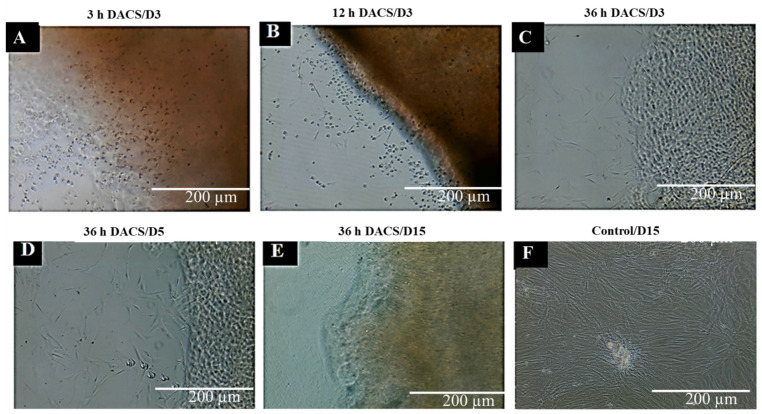
Microphotographs of the relative cytotoxicity of DACS scaffold on chondrocytes via phase-contrast microphotography. (**A**) 3-h decellularized DACS reseeded with Chondrocytes and cultured for 24 h. (**B**) 12-h decellularized DACS reseeded with Chondrocytes and cultured for 24 h. (**C**) 36-h decellularized DACS reseeded with Chondrocytes and cultured for 24 h. (**D**) 36-h decellularized DACS reseeded with Chondrocytes and cultured for 5 days. (**E**) 36-h decellularized DACS reseeded with Chondrocytes and cultured for 15 days. (**F**) Monolayer plate control chondrocytes after 15 days of culturing.

**Figure 12 materials-15-01974-f012:**
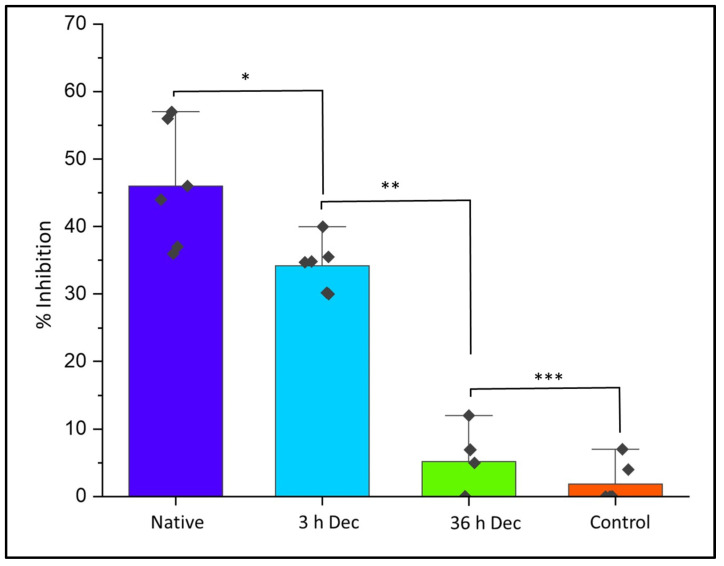
Assessment of the cytotoxic inhibition of chondrocytes by DACS scaffolds after various stages of decellularization. *n* = 6, * *p*-value = 0.00415, ** *p*-value = 1.19845 × 10^−8^, *** *p*-value = 0.1998.

**Figure 13 materials-15-01974-f013:**
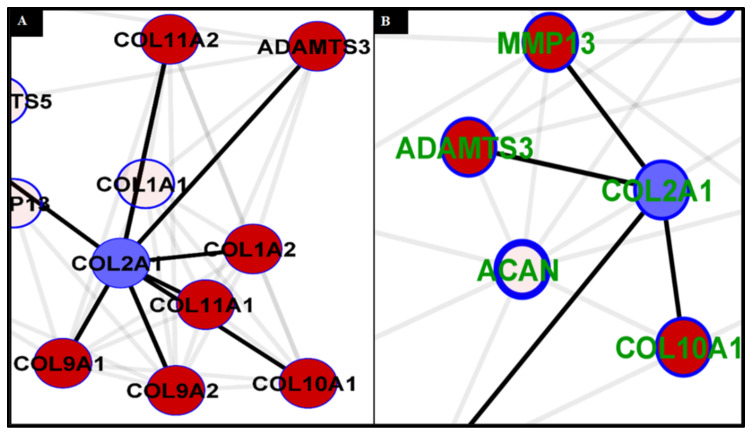
(**A**) String Network of Col2A1, showing the protein’s interaction with other collagen molecules and neighboring molecules in a normal cartilage tissue formation process. Direct interactions are indicated by the red color, whereas neighboring molecules are empty circles. (**B**) String Network of Col2A1, showing the protein’s interaction with other collagen molecules and neighboring molecules in abnormal cartilage formation or cartilage-related disease. Direct interactions are indicated by the red color, whereas neighboring molecules are empty circles. Col2A1-Network (String: PubMed query) created by Cytoscape 3.8.2. and data imported from the curated public databases with the target species being Homo sapiens only.

**Figure 14 materials-15-01974-f014:**
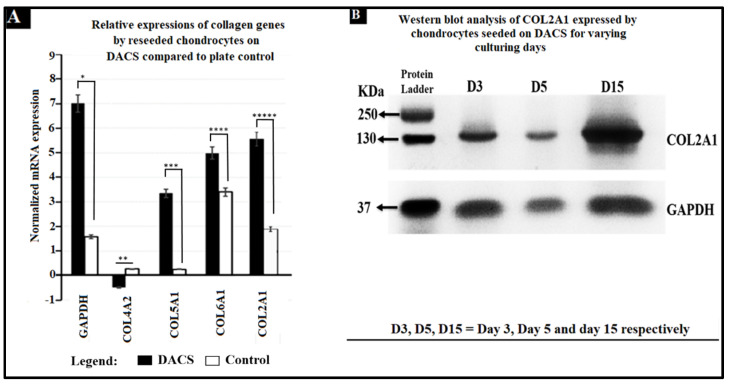
Relative expression of collagen genes of chondrocytes seeded on DACS scaffold compared to the plate control and western blot analysis of Col2A1 of chondrocytes seeded on DACS construct. (**A**) Collagen 2A1, 5A1, 6A1 up-regulated in chondrocytes grown in 3D DACS scaffold, and Collagen 4A2 was observed to be downregulated. (**B**) Shows the western blot analysis of Col2A1 protein expressed by chondrocytes seeded on DACS scaffold at various days of culturing. These molecules drive chondrocytes remodeling of scaffold ECM, ECM formation and deposition, and play a crucial role in adhesion. Relative expression of collagen genes of chondrocytes seeded on DACS scaffold compared to the plate control. Data were pooled from three independent experiments, and error bars show mean ± SEM. Significance was determined by Student *t*-test, two tails, with *p* < 0.01 determined for all. N = 3. * *p*-value < 0.001, ** *p*-value = 0.0524, *** *p*-value < 0.001, **** *p*-value = 0.0046, ***** *p*-value < 0.001.

**Figure 15 materials-15-01974-f015:**
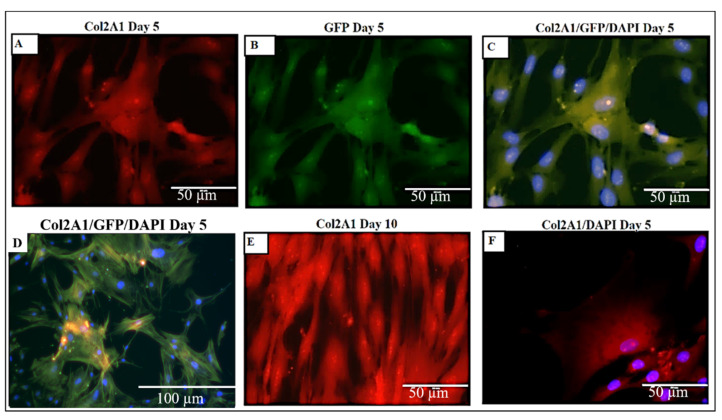
Progressive accumulation of Col2A1 in the cytoplasm of chondrocytes grown on DACS as culturing days increases. Col2A1 (Red) expressed on day 5 in cells seeded on DACS (probed with anti-Col2A1) (**A**,**E**). Cells on DACS stained with Phalloidin (as shown in green, **B**). Phalloidin stained merged with Col2A1 and DAPI (**C**). (**D**) Cells on monolayer control probed for Col2A1 (red), stained with phalloidin (green), and DAPI (blue). (**F**) Col2A1 expressed in chondrocytes in control monolayer merged with DAPI day 5 of culturing. Scale bar, 50 µm for **A**–**F**. Scale bar, 100 µm for **D**.

**Table 1 materials-15-01974-t001:** Primers used for real time QPCR analysis.

Name	Sequence	Input Template ID
FW COL2A1	5′-GTAGAGACCCGGACCCGC-3′	NM_001844.5
RV COL2A1	5′-ACTCTCCGAAGGGGATCTCA-3′	NM_001844.5
FW COL5A1	5′-TTCAAGCGTGGGAAACTGCT-3′	NM_000093.5
RV COL5A1	5′-GGGAGAAGCCTTCACTGTCC-3′	NM_000093.5
FW COL4A2	5′-ATAGGAGGGCCCAAGGGATT-3′	NM_001846.4
RV COL4A2	5′-CAGGGTCCCCTCTATCACCA-3′	NM_001846.4
FW COL6A1	5′-ATTGCCAAGGACTTCGTCGT-3′	NM_001848.3
RV COL6A1	5′-ACATTGAGCTGGTCTGAGCC-3′	NM_001848.3
FW GAPDH	5′-CCATGGGGAAGGTGAAGGTC-3′	NM_002046.7
RV GAPDH	5′-AGTGATGGCATGGACTGTGG-3′	NM_002046.7

## Data Availability

Data is contained within the article.

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
