# Peer review of "Decellularized Avian Cartilage, a Promising Alternative for Human Cartilage Tissue Regeneration"

_materials, 2022, doi:10.3390/ma15051974_

Round 1
Reviewer 1 Report
General comments to the Authors:
This group has developed a strategy to make porous decellularized matrices from Avian Articular Cartilage to support and regenerate cartilage tissue. Although common chemicals such as SDS and Triton X were employed to produce decellularized tissue scaffolds (DACS), the significance of this work is extensive characterization on the decellularized scaffold to confirm the absence of immunogenic components. In addition, this chemical process preserves the collagen and glycosaminoglycan, which are essential to provide physical, mechanical and biochemical support or signals to the growing cells. Developed decellularized matrices were biocompatible and were non-toxic to the human chondrocytes. Overall, this study is warrant to publish with minor revision.
Specific comments to the Authors:
- While the use of animal tissues, the main concern is the possibility of transmission of zoonotic diseases. The rationale for the use of avian cartilage tissue is not clear. Also, it is warrant to discuss zoonotic disease transmission clearly in the manuscript.
- Was there any reason for frozen the tissues before decellularization? Why was the tissue frozen? What temperature is used? It is necessary to mention the appropriate citation for this decellularization procedure.
- What is the size and thickness of the DACS used for cell culture studies?
- Compare and discuss the efficiency of DACS with other forms of decellularized matrices used in cartilage regeneration. Is this DACS superior or inferior?
Author Response
- While the use of animal tissues, the main concern is the possibility of transmission of zoonotic diseases. The rationale for the use of avian cartilage tissue is not clear. Also, it is warrant to discuss zoonotic disease transmission clearly in the manuscript.
We will first and foremost thank the reviewer for the thorough review of the manuscript and the weakness pointed out to us. We have revised our manuscript to include all comments made by the reviewer.
One major setback in tissue graft or organ transplant from a donor of a different species from that of the recipient is the possibility of zoonosis and immunological rejection, this has been briefly discussed in the introduction section of this work, and shows that decellularization is able to overcome such limitations. We appreciate the reviewer for this pointer.
- Was there any reason for frozen the tissues before decellularization? Why was the tissue frozen? What temperature is used? It is necessary to mention the appropriate citation for this decellularization procedure.
The cartilage tissue was frozen first in order to carry out the thin section of the tissue using the microtome. More so, the freeze-thaw process disrupts the chondrocytes embedded in the thick cartilage and enhances quicker decellularization.
- What is the size and thickness of the DACS used for cell culture studies?
The dimensions of the sections used for the study were 2 mm by 5mm by 0.5mm for width, length and thickness respectively. We have updated this in the manuscript.
- Compare and discuss the efficiency of DACS with other forms of decellularized matrices used in cartilage regeneration. Is this DACS superior or inferior?
We have briefly discussed and compared DACS with other decellularized scaffolds for cartilage regeneration. We thank reviewer for this input, it has improved the manuscript.

Reviewer 2 Report
Based on the results of an in-depth evaluation that I have done for an article with the title “Decellularized avian cartilage, a promising alternative for human cartilage tissue regeneration” (Advances in Biomaterials), I think this article should be rejected for publication in Materials or reconsider after proper changes in major revision.
- I would encourage and advise the authors to adopt some of the additional references in the introduction section published by MDPI as follow:
- Tresca Stress Simulation of Metal-on-Metal Total Hip Arthroplasty during Normal Walking Activity. Materials (Basel). 2021, 14, 7554. https://doi.org/10.3390/ma14247554
- The Effect of Bottom Profile Dimples on the Femoral Head on Wear in Metal-on-Metal Total Hip Arthroplasty. J. Funct. Biomater. 2021, 12, 38. https://doi.org/10.3390/jfb12020038
- To improve the quality of English used in this manuscript and make sure English language, grammar, punctuation, spelling, and overall style are correct, further proofreading is needed. As an alternative, the authors can use the MDPI English proofreading service for this issue.
- Describe the novelty of the article made by the author? From the results of my evaluation, it seems that many similar published works adequately explain what you have raised in the current manuscript. If there is something really new in this manuscript, please highlight it more clearly in the introduction. This is the fundamental reason why this manuscript needs a major revision or should be rejected.
- The author must provide a brief specification regarding all tools used in the research carried out so that the reader can estimate the accuracy and differences in the results that the authors describe due to the use of different tools in future studies.
- In the result and discussion section, if possible, the authors should compare the result in the present study with the published literature in the similar/identical condition and explain this comparison become comprehensive discussion to ensure the result obtained in the present study.
- In the results and discussion section, the author must describe the limitation of the research carried out.
- The conclusion of this manuscript is not solid. Further elaboration is needed.
- Further research needs to be explained in the conclusion section.
- I would encourage and advise the authors to put and discuss the figure from the supplementary file in the main text of the manuscript. Please do not put it in the supplementary.
Author Response
- I would encourage and advise the authors to adopt some of the additional references in the introduction section published by MDPI as follow:
- Tresca Stress Simulation of Metal-on-Metal Total Hip Arthroplasty during Normal Walking Activity. Materials (Basel). 2021, 14, 7554. https://doi.org/10.3390/ma14247554
- The Effect of Bottom Profile Dimples on the Femoral Head on Wear in Metal-on-Metal Total Hip Arthroplasty. J. Funct. Biomater. 2021, 12, 38. https://doi.org/10.3390/jfb12020038
References incorporated into manuscript.
- To improve the quality of English used in this manuscript and make sure English language, grammar, punctuation, spelling, and overall style are correct, further proofreading is needed. As an alternative, the authors can use the MDPI English proofreading service for this issue.
Responses. Thank manuscript has been subjected to Grammarly check for English/grammer corrections.
- Describe the novelty of the article made by the author? From the results of my evaluation, it seems that many similar published works adequately explain what you have raised in the current manuscript. If there is something really new in this manuscript, please highlight it more clearly in the introduction. This is the fundamental reason why this manuscript needs a major revision or should be rejected.
Novelty of work has been highlighted.
- The author must provide a brief specification regarding all tools used in the research carried out so that the reader can estimate the accuracy and differences in the results that the authors describe due to the use of different tools in future studies.
Tools used for the study has been specified.
- In the result and discussion section, if possible, the authors should compare the result in the present study with the published literature in the similar/identical condition and explain this comparison become comprehensive discussion to ensure the result obtained in the present study.
Comparison has been made with various scaffolds (both artificial and natural base scaffolds).
- In the results and discussion section, the author must describe the limitation of the research carried out.
Limitation of the work has been highlighted in the discussion.
- The conclusion of this manuscript is not solid. Further elaboration is needed.
Conclusion of the manuscript has been revised.
- Further research needs to be explained in the conclusion section.
“Further research” has been explained.
- I would encourage and advise the authors to put and discuss the figure from the supplementary file in the main text of the manuscript. Please do not put it in the supplementary.
Supplementary figures are no incorporated into the text as figures and no more supplementary.
Reviewer 3 Report
In this work, the authors reported scaffolds made from decellularized avian cartilage with potential for human cartilage regeneration and repair. Survival and proliferation of human chondrocytes on these decellularized cartilages were carefully tested. Overall, the work is fitting to Materials, whereas a few minor issues should be addressed before acceptance of publication.
- Considering the similarity, accessibility and lower cost of chicken cartilage, is there any reason the authors chose avian cartilage rather than chicken?
- The authors should better provide scanning electron microscopy images of decellularized avian cartilages.
- There might be conflicts in terms of P values in Figure 2b, 3a, and 8. For instance, in figure 2b, the SDS Dec 12hrs group seemed to be far lower than 3hr group whereas somehow its P value in comparison to Native group was higher than that of 3hr to Native.
- It looks like decellularized avian cartilage had good performance to support tissue regeneration. How about the superiorities of these materials in comparison with well-recognized artificial cartilages or commercial products?
- Figures of control group should be displayed in Figure 6.
Author Response
In this work, the authors reported scaffolds made from decellularized avian cartilage with potential for human cartilage regeneration and repair. Survival and proliferation of human chondrocytes on these decellularized cartilages were carefully tested. Overall, the work is fitting to Materials, whereas a few minor issues should be addressed before acceptance of publication.
- Considering the similarity, accessibility and lower cost of chicken cartilage, is there any reason the authors chose avian cartilage rather than chicken?
Manuscript has been revised to indicate specifically that the given Avian cartilage used was from Gallus Gallus domesticus (chicken).
- The authors should better provide scanning electron microscopy images of decellularized avian cartilages.
Manuscript has been revised so that SEM of decellularized cartilage has been provided.
- There might be conflicts in terms of P values in Figure 2b, 3a, and 8. For instance, in figure 2b, the SDS Dec 12hrs group seemed to be far lower than 3hr group whereas somehow its P value in comparison to Native group was higher than that of 3hr to Native.
Manuscript has been revised to show figures having pairwise comparison, the data was evaluated by an independent researcher and p-values produced the same values using the same paired sample t-test from Origin Pro software.
- It looks like decellularized avian cartilage had good performance to support tissue regeneration. How about the superiorities of these materials in comparison with well-recognized artificial cartilages or commercial products?
Comparison with chondro-Gide, a well-recognized artificial and commercial cartilage regeneration scaffold, has been made in the revised manuscript.
- Figures of control group should be displayed in Figure 6.
Figures of control group has been added in to Figure 6.
Reviewer 4 Report
The work is of great interest, as it is devoted to the development of artificial organs based on decellularized cartilage of an avian. A detailed analysis of the disposal of cartilage from cellular elements, proteins, fats, nucleic acids, which are capable of initiating an immune response and rejection of implanted cartilage plates, was carried out.
There are a number of weaknesses of this study:
1) the problem of the presence of antibacterial drugs and growth hormones in the bones of birds has not been investigated. It is known that in poultry farms the addition of antibiotics is a prerequisite in view of the need to reduce deaths among birds. Also, some factories use growth hormones to increase the proportion of the muscle frame in birds. Antibiotics and hormones can inhibit stem cells.
2) A huge disadvantage of the work is not the ability to work with a statistical software package and, as a result, the poor quality of the description of the methods of checking the data obtained for compliance with the normality of the distribution of signs, and as a consequence of the choice of the method of representing the average and the tendency of scattering of signs. This, in turn, leads to the choice of assessing the statistical significance of differences between groups. So the authors use the data representation as the average and the error of the average, which does not fully reflect the dispersion (spread) of the studied features - the data should be given as the average and standard deviation. Further, the authors state that they used a pairwise data comparison method to identify statistically significant differences. But the figures show that some data have a comparison between more than 2 groups. There are no studies on the presence of correlations between the functional properties of chondrocytes and the type of decellurization, the timing of decellularization of cartilage.
3) There are parentheses in the text, in which there is nothing, either the authors forgot to add something or it is a typo.
4) It is advisable to give a transcript of the abbreviation at the first time.
Author Response
The work is of great interest, as it is devoted to the development of artificial organs based on decellularized cartilage of an avian. A detailed analysis of the disposal of cartilage from cellular elements, proteins, fats, nucleic acids, which are capable of initiating an immune response and rejection of implanted cartilage plates, was carried out.
There are a number of weaknesses of this study:
1) the problem of the presence of antibacterial drugs and growth hormones in the bones of birds has not been investigated. It is known that in poultry farms the addition of antibiotics is a prerequisite in view of the need to reduce deaths among birds. Also, some factories use growth hormones to increase the proportion of the muscle frame in birds. Antibiotics and hormones can inhibit stem cells.
This comment has been attended to in line 65 to 77. We thank the reviewer for this crucial insight.
2) A huge disadvantage of the work is not the ability to work with a statistical software package and, as a result, the poor quality of the description of the methods of checking the data obtained for compliance with the normality of the distribution of signs, and as a consequence of the choice of the method of representing the average and the tendency of scattering of signs. This in turn, leads to the choice of assessing the statistical significance of differences between groups. So the authors use the data representation as the average and the error of the average, which does not fully reflect the dispersion (spread) of the studied features - the data should be given as the average and standard deviation. Further, the authors state that they used a pairwise data comparison method to identify statistically significant differences. But the figures show that some data have a comparison between more than 2 groups. There are no studies on the presence of correlations between the functional properties of chondrocytes and the type of decellularization, the timing of decellularization of cartilage.
Comparisons were made between two groups; this figures have been revised to show these comparisons only. The functional properties of chondrocytes and type of decellularization will be our next interesting venture, however we showed the biocompatibility of the decellularized constructs with respect to timing of decellularization which showed that after 36 h of decellularization, the constructs were biocompatible to seeded chondrocytes.
3) There are parentheses in the text, in which there is nothing, either the authors forgot to add something or it is a typo.
Parentheses has been removed from text. Thank you.
4) It is advisable to give a transcript of the abbreviation at the first time.
Manuscript has been revised to have abbreviations described at their first appearance.
Reviewer 5 Report
This research paper by Professor Dean and Ayariga is well-crafted and detailed. The authors analyze the different aspects of qualifying an exogenous implant material (in this case, Decellularized Avian Cartilage scaffolds) for cartilage regeneration applications. Their methodology to test the physico-chemical as well as biological aspects necessary for chondrocytes to instigate tissue repair are well-organized. The figures are neat, and easy to understand. There are some minor changes that I would recommend for adding perspective on this aspect:
- Around line 65, the authors can speak to the details of foreign body response and implant acceptance. It is important to prove some clarity on the properties of exogenous materials required for healthy implant acceptance. Here, they can cite:
- Anderson JM, Rodriguez A, Chang DT. Foreign body reaction to biomaterials. Semin Immunol. 2008 Apr;20(2):86-100. doi: 10.1016/j.smim.2007.11.004.
- Wei F, Liu S, Chen M, et al. Host Response to Biomaterials for Cartilage Tissue Engineering: Key to Remodeling. Front Bioeng Biotechnol. 2021;9:664592.
- Boddupalli A, Zhu L, Bratlie KM. Methods for Implant Acceptance and Wound Healing: Material Selection and Implant Location Modulate Macrophage and Fibroblast Phenotypes. Adv Healthc Mater. 2016 Oct;5(20):2575-2594.
- Around line 445, there should be more detail provided of what would be a healthy collagen organization response in terms of collagen ratios and structure for appropriate cartilage regeneration.
- On line 456 -- there is an empty bracket that needs to be either filled in for citation or removed
Author Response
This research paper by Professor Dean and Ayariga is well-crafted and detailed. The authors analyze the different aspects of qualifying an exogenous implant material (in this case, Decellularized Avian Cartilage scaffolds) for cartilage regeneration applications. Their methodology to test the physico-chemical as well as biological aspects necessary for chondrocytes to instigate tissue repair are well-organized. The figures are neat, and easy to understand. There are some minor changes that I would recommend for adding perspective on this aspect:
- Around line 65, the authors can speak to the details of foreign body response and implant acceptance. It is important to prove some clarity on the properties of exogenous materials required for healthy implant acceptance. Here, they can cite:
- Anderson JM, Rodriguez A, Chang DT. Foreign body reaction to biomaterials. Semin Immunol. 2008 Apr;20(2):86-100. doi: 10.1016/j.smim.2007.11.004.
- Wei F, Liu S, Chen M, et al. Host Response to Biomaterials for Cartilage Tissue Engineering: Key to Remodeling. Front Bioeng Biotechnol. 2021;9:664592.
- Boddupalli A, Zhu L, Bratlie KM. Methods for Implant Acceptance and Wound Healing: Material Selection and Implant Location Modulate Macrophage and Fibroblast Phenotypes. Adv Healthc Mater. 2016 Oct;5(20):2575-2594.
We thank the reviewer to these crucial additions. These comments have been added and properly referenced.
- Around line 445, there should be more detail provided of what would be a healthy collagen organization response in terms of collagen ratios and structure for appropriate cartilage regeneration.
Description of the collagen alignment and amounts are discussed.
- On line 456 -- there is an empty bracket that needs to be either filled in for citation or removed
Empty bracket removed.
Round 2
Reviewer 2 Report
Dear Ayariga et al.,
After carefully reading the author's revised manuscript entitled "Decellularized avian cartilage, a promising alternative for human cartilage tissue regeneration" (materials-1601279) by Ayariga et al., The authors have been made significant improvements in the revised manuscript. Also, all of the issue in my review report has been addressed precisely.
With my pleasure, I recommend the manuscript should be accepted for publication on Materials.
Best regards,
The Reviewer